# Exploring geometrical stereoscopic aerosol top height retrieval from geostationary satellite imagery in East Asia

Minseok Kim[1], Jhoon Kim[1], Hyunkwang Lim[2], Seoyoung Lee[1], Yeseul Cho[1], Huidong Yeo[3], and Sang-Woo Kim[3]

[1]Department of Atmospheric Sciences, Yonsei University, Seoul, Korea
[2]National Institute for Environmental Studies, Tsukuba, Japan
[3]School of Earth and Environmental Sciences, Seoul National University, Seoul, Korea

*Correspondence to*: Jhoon Kim (jkim2@yonsei.ac.kr)

**Abstract.** Despite the importance of aerosol height information for events such as volcanic eruptions and long-range aerosol transport, spatial coverage of its retrieval is often limited because of a lack of appropriate instruments and algorithms. Especially, geostationary satellite observations provide constant monitoring for such events. This study assessed the application of different viewing geometries for a pair of geostationary imagers to retrieve aerosol top height (ATH) information. The stereoscopic algorithm converts the lofted aerosol layer parallax, calculated using image-matching of two visible images, to ATH. The sensitivity study provides a reliable result using a pair of Advanced Himawari Imager (AHI) and Advanced Geostationary Radiation Imager (AGRI) images at 40° longitudinal separation. The pair resolved aerosol layers above 1 km altitude over East Asia. In contrast, aerosol layers must be above 3 km to be resolved by paired AHI and Advanced Meteorological Imager (AMI) images at 12.5° longitudinal separation. Case studies indicate that the stereoscopic ATH retrieval results are consistent with aerosol heights determined using extinction profiles from the Cloud–Aerosol Lidar with Orthogonal Polarization (CALIOP). Comparisons between the stereoscopic ATH and the CALIOP 90 % extinction height, defined by extinction coefficient at 532 nm data, indicated that 71.3 % of ATH estimates from the AHI and AGRI are within 2 km of CALIOP 90 % extinction heights, compared with 49.3 % from the AHI and AMI. The ability of the stereoscopic algorithm to monitor hourly aerosol height variations is demonstrated by comparison with a Korea Aerosol Lidar Observation Network dataset.

## 1 Introduction

Long-range transboundary transport of aerosols from wildfires and deserts affects air quality over wide areas. Information on aerosol height is crucial in determining the effects of lofted aerosol plumes transported to distant downwind regions. Lidar systems provide detailed vertical profiles of aerosol layers and are primary tools in providing vertically resolved information. Ground-based lidars are capable of high-quality aerosol profiling with little interference from surface signals, and they have been used for long-term analysis of aerosol vertical distributions (Tian et al., 2017; Gupta et al., 2021). However, the requirement for manual maintenance means lidar sites are concentrated in urban areas. Spaceborne lidar instruments such as

Cloud–Aerosol Lidar with Orthogonal Polarization (CALIOP; Winker et al., 2009; Wu et al., 2016) provide information on the vertical distributions of aerosol layers worldwide (Kim et al., 2018). Previous studies have used CALIOP observation products to demonstrate seasonal variability in the vertical structures of aerosols over China (Huang et al., 2013; Liao et al., 2021). However, active sensors such as CALIOP have narrow swaths (e.g., CALIOP footprint diameter is 70 m; Winker et

al., 2010), which means that they may miss aerosol transport events most of the time. Even if the spaceborne lidar system passes over the aerosol layer, the horizontal variability of the layer outside the swath remains unknown.

Aerosol height retrieval algorithms using passive sensors have been developed to meet the need for a better understanding of aerosol vertical distribution over broader areas. Choi et al. (2021) determined information content requirements for passive remote-sensing measurements when profiling aerosols below the planetary boundary layer, using oxygen ($O_2$) A and B bands

with higher spectral resolution and polarimetric measurements. For this method, conventional spectroscopic aerosol height retrieval algorithms make use of the absorption bands of $O_2$, which normally have a stable vertical distribution. Scattered photons travel longer atmospheric paths in aerosol layers at lower altitudes, leading to greater absorption by $O_2$ and less backscattered radiance being observed from satellite observations. An aerosol height retrieval algorithm using the TROPOspheric Monitoring Instrument (TROPOMI), a hyperspectral spectrometer on-board the Sentinel-5 Precursor

(Sentinel-5P), was developed and validated by Nanda et al. (2020), where aerosol height retrieval using TROPOMI utilized hyperspectral observations of the $O_2$ A band. Aside from the underestimation induced by bright surfaces on land, the results agreed well with CALIOP weighted extinction heights. With additional use of the $O_2$ B band, Chen et al. (2021) developed an aerosol height retrieval mechanism for TROPOMI for application in surface particulate matter estimation, with the algorithm being applied to absorbing aerosols such as dust and smoke. Additional use of the $O_2$ B band yields systematically

higher (by ~1.6 km) aerosol optical central heights than the TROPOMI operational aerosol height, nearer the height with the strongest backscatter for CALIOP. $O_2$-$O_2$ band was also used to retrieve aerosol layer height (Park et al., 2016; Chimot et al., 2017). A spectroscopic aerosol layer height retrieval using single passive geostationary imager over ocean has been developed and validated during Korea-United States Air Quality Study campaign period (KORUS-AQ) (Lim et al., 2023).

Studies have shown that the use of geometrical features of elevated atmospheric structures apparent to multiple sensor

imagery is effective, rather than using computationally expensive radiative transfer calculations. Using stereography, unlike spectroscopic algorithms, one gets to retrieve feature top height. Aerosol plume height retrievals by the Multi-angle Imaging SpectroRadiometer (MISR) Interactive eXplorer (MINX) system make use of multi-angle imagery from nine push-broom cameras on-board the Terra satellite (Nelson et al., 2013). These cameras view 360 km swaths of Earth, with nine viewing angles ranging from nadir to 70.5°. MINX successfully verified aerosol height retrieval using stereoscopic imagery with

passive sensors. The synergetic use of two or more sensors for aerosol height retrieval has also been studied (Chu et al., 2008; Lee et al., 2015; Lee et al., 2021). Despite the use of spaceborne passive sensor observations improving the spatial coverage of aerosol vertical structure information, monitoring of diurnal variations in aerosol layers is restricted by the low temporal resolution of low-Earth-orbit (LEO) satellites.

Observation of aerosol vertical structures using geostationary Earth orbit (GEO) satellites has potential for continuously monitoring diurnal variations in aerosol transport over broad areas. For the retrieval of aerosol optical depth (AOD), the use of geostationary meteorological satellites such as the Meteorological Imager (MI), Advanced Baseline Imager (ABI), and Advanced Himawari Imager (AHI) to obtain aerosol optical properties has been well established and their proficiency has been demonstrated (Kim et al., 2008; Kim et al., 2016; Lim et al., 2018, Zhang et al., 2020). However, the visible to infrared (VIS–IR) wavelength channels that are usually employed by meteorological satellite instruments lack sensitivity to aerosol height information, thus insufficient for the retrieval of aerosol height from observed radiances. Geostationary hyperspectral spectroscopy employed by the Geostationary Environment Monitoring Spectrometer (GEMS), Tropospheric Emissions: Monitoring of Pollution (TEMPO), and Sentinel-4 is expected to offer new possibilities for diurnal monitoring of aerosol height (Ingmann et al., 2012; Kim et al., 2020; Zoogman et al., 2017). The optimal estimation-based algorithm for aerosol height retrieval using the GEMS on-board GeoKompsat-2B (GK-2B) was developed using proxy data from the Ozone Monitoring Instrument (OMI) and is to be applied to operational GEMS observations (Kim et al., 2018).

In this study, we explore a geometrical method using VIS observations from two geostationary meteorological satellites to retrieve aerosol top height (ATH). Cloud top heights have been successfully retrieved using geometrical fusion of two LEO/GEO satellite images (Hasler, 1981; Seiz et al., 2007; Zašek et al., 2013; Merucci et al., 2016; Lee at al., 2020), suggesting the applicability of such a method to any structures in the atmosphere. However, typical aerosol layers are formed due to surface pollution emission in East Asia thus are not as optically thick as clouds or volcanic ash plumes. Also, aerosol layers tend to be at a much lower height than cloud or volcanic ash plumes. So, the applicability and accuracy of the geometrical method for estimating aerosol feature height needs to be investigated. Our aims were to investigate the sensitivity of ATH to observed parallax, and to validate our results by comparison with other aerosol profile datasets. The datasets used are described in Section 2. Section 3 introduces a stereoscopic ATH retrieval algorithm, and Section 4 assesses the sensitivity and uncertainty of the height retrieval algorithm based on pairs of sensors. Section 5 discusses the ATH results and compares them with spaceborne and ground-based lidar data. Finally, Section 6 summarizes the skill of the algorithm and suggests prospects for future studies.

## 2 Data

### 2.1 Satellite observation data

#### 2.1.1 Advanced Himawari Imager

The AHI is a meteorological instrument on-board the Japanese satellites Himawari-8 and -9, which were launched on 7 October 2014 and 2 November 2016, respectively. The AHI continues the mission of the Multi-Functional Transport Satellite (MTSAT) with enhanced specifications of 16 spectral bands, including 3 VIS, 1 near-IR (NIR), 2 shortwave IR, and 10 IR channels. Two of the VIS bands have spatial resolutions at the sub-satellite point of 1 km (blue, green; 470, 510 nm),

and the other is at 0.5 km resolution (red; 640 nm). Full-disk AHI scans are conducted every 10 min. The Image Navigation and Registration (INR) variation for AHI is less than 0.5 pixels for VIS bands, or ~500 m for the 1 km resolution bands (blue, green) and 250 m for the 500 m resolution band (red) (Tabata et al., 2016).

The AHI AOD data from the Yonsei AErosol Retrieval (YAER) algorithm are used as a criterion for the selection of retrieval pixels. The YAER algorithm is a multi-channel algorithm based on three VIS and one NIR AHI channels. The product demonstrates good agreement with Aerosol Robotic Network (AERONET) AOD data (Lim et al., 2018).

### 2.1.2 Advanced Meteorological Imager

The Advanced Meteorological Imager (AMI) is a GEO meteorological instrument on-board Geo-Kompsat 2A (GK-2A), which was launched on 4 December 2018 by the National Meteorological Satellite Center (NMSC) of Korea succeeding the mission of its MI predecessor. The AMI spectral bands are similar to those of AHI. AMI also has 16 spectral bands, including 3 VIS, 1 NIR, 2 shortwave IR, and 10 IR channels. Blue and green bands (470, 510 nm) have spatial resolutions at the sub-satellite point of 1 km, and a red band (640 nm) has a 0.5 km resolution. The AMI also carries out full-disk scans every 10 min. An AMI INR evaluation on 31 July 2019 indicated an absolute navigation error of <1.0 pixel (~1 km) for 1 km resolution bands and 0.9 pixels (~450 m) for 500 m bands. The required INR performance was 1.50 km for 1 km resolution bands, and 0.75 km for 500 m resolution band (Kim et al., 2021). The NMSC reports the navigation performance of GK-2A on a regular basis (https://nmsc.kma.go.kr/homepage/html/satellite/quality/selectQualityGk2a.do, last access: 29 March 2023), with monthly navigation performance reports from 2020 indicating average navigation errors of 9–13 and 8–18 µrad for the latitudinal (N–S) and longitudinal (E–W) directions, respectively. Aside from a few cases of extreme INR error, navigation errors in either direction have typically fluctuated by <20 µrad since May 2020, whereas those of the N–S direction before April 2020 frequently exceeded 20 µrad (although they were usually <40 µrad).

### 2.1.3 Advanced Geostationary Radiation Imager

The Chinese geostationary satellite Feng-Yun-4A (FY-4A), which was launched on 11 December 2016, carries the Advanced Geostationary Radiation Imager (AGRI), the Geostationary Interferometric Infrared Sounder (GIIRS), and the Lightning Mapping Imager (LMI). The AGRI observes 14 VIS–IR spectral bands, but with only two VIS bands of blue and red (470 and 680 nm) that have spatial resolutions of 1 and 0.5 km, respectively. AGRI performs 40 full-disk scans per day, using observations from every hour to match the times to the other sensors. The INR requirement for AGRI is 112 µrad or ~4 km on the surface (Yang et al., 2017).

### 2.2 Data intercomparison

Through intercomparison with aerosol profile data from lidars, spaceborne aerosol height retrieval algorithms with passive sensors can be evaluated. Spaceborne lidars such as CALIOP enable evaluation of aerosol height over broad spatial ranges

and areas where ground-based lidars are unavailable. Several studies have thus used CALIOP data to evaluate their algorithms (Lee et al., 2015; Nanda et al., 2020; Chen et al., 2021; Lee et al., 2021). Despite their shortcoming in terms of sparse observational coverage, ground-based lidars facilitate the monitoring of diurnal variations in aerosol height. In this study, we used both spaceborne and ground-based lidar data for comparisons among stereoscopic ATH algorithm products.

### 2.2.1 CALIOP

The Cloud–Aerosol Lidar and Infrared Pathfinder Satellite Observations (CALIPSO) satellite was launched on 28 April 2006 on a sun-synchronous orbit, and it revisits the same ground track every 16 days. CALIOP, the primary instrument on-board CALIPSO, is a two-wavelength polarization lidar optimized for global profiling of cloud and aerosol distributions. CALIOP measures the returning signals of pulses at 532 and 1064 nm that are produced simultaneously by the laser. Two receivers of the 532 nm channel, both of which can detect polarization signals, measure the degree of linear polarization of

the returning signal. Using the signals of the two 532 nm channels and a total 1064 nm returning signal, CALIOP obtains accurate information on cloud and aerosol height (Winker et al., 2007). The standard output aerosol profile product includes total, parallel, and perpendicular backscatter at 532 nm, extinction coefficients (ECs), particulate depolarization ratios at 532 nm, and volume backscatter and ECs derived from the 1064 nm channel. Data are reported at a uniform spatial resolution of 60 m (vertical) and 5 km (horizontal). Vertical resolutions are coarser at higher altitudes because of CALIPSO's onboard

data-averaging scheme. For the sake of data quality, we used aerosol extinction data satisfying cloud–aerosol discrimination (CAD) scores of −20 to −100 (with CAD scores closer to −100 having high confidence) and extinction quality control flags of 1 and 2 at 532 nm.

### 2.2.2 Korean Aerosol Lidar Observation Network

Aerosol extinction profile data from ground-based lidars provide quantitative information on aerosol vertical distributions.

Without surface interference, an aerosol extinction profile can be obtained from the returning signal of a ground-based lidar using the lidar equation (Welton et al., 2000). The lidar ratio, which depends on aerosol type, must be assumed from previous studies or sun-photometer observations near the lidar site. Attenuation of a thick aerosol layer by low-level cloud decreases the signal-to-noise ratio, preventing aerosol extinction profile retrieval. The Korean Aerosol LIDAR Observation Network (KALION) is a network of aerosol lidars providing real-time monitoring of aerosol formation and transport over

Korea. Lidar observation images at each site in Korea are updated daily on the KALION website (http://www.kalion.kr, last access: 29 March 2023). Six lidar sites operated by five institutes on the Korean Peninsula undertake continuous observations of aerosol formation and transport. Here, we used total attenuated backscatter data from lidar sites in Seoul (SNU) and Gosan (GSN), maintained by Seoul National University (Yeo et al., 2016), for comparison with results from the stereoscopic ATH algorithm. KALION total attenuated backscatter profiles have a vertical resolution of 60 m and temporal

resolution of 15 min.

# 3 Stereoscopic ATH algorithm

## 3.1 Overview

A flowchart of the stereoscopic ATH algorithm is shown in Fig. 1. It begins with a resampling procedure, bringing one geolocation coordinate to the other. The AHI geolocation serves as the reference for fixing the top-of-atmosphere (TOA) reflectance images of the AMI or AGRI. TOA reflectance data within the VIS-NIR range can be used for retrieval, as aerosols are optically visible in these bands. However, the land surface is brighter at longer wavelengths, while the ocean surface is brighter at shorter wavelengths (von Hoyningen-Huene et al., 2010). Therefore, the blue (470 nm) and green (510 nm) bands suffer from interference from ocean surface signals, and the NIR band (850 nm) from land surface signals. To avoid errors induced by surface signals, we used red band (640 nm) TOA reflectance for both land and ocean.

Parallax is defined as the effect by which the position of an object appears to change when viewed from different positions. When an aerosol layer is observed from different GEO orbits, it appears to be above different points on Earth. The higher the aerosol layer is, the longer the parallax. After resampling, the parallax is calculated with an image-matching process. Image-matching identifies identical aerosol layers in each satellite image. Then, the parallax can be calculated by measuring the distance between the two points on Earth over which the aerosol layer appears to be located. Finally, a simple 3-dimensional (3D) parallax height conversion equation was defined to determine the height of the aerosol layer. The retrieved product is defined as ATH measured from the surface of a spherical Earth with a radius of 6378.2 km. In Fig. 2, scanning points on the ground (A' and B') from satellites A and B are projected onto the sphere above. As the distance between two scanning points (parallax) is close enough to assume a spherical Earth, the ATH converted from the parallax is the altitude from the surface of the sphere with a radius of 6378.2 km.

The stereoscopic ATH algorithm is developed based on the concept of cloud top height retrieval methods (Lee et al., 2020). However, unlike clouds, aerosol layers are optically thin. This implies that even when using the same concept of algorithm, the height that the stereoscopic ATH algorithm gives can vary in different situations. As mentioned above, parallax is calculated by finding identical aerosol layers from satellite images. This works best when the aerosol layer is dense enough to screen the surface signals. In this case, the algorithm is likely to give the height near the top of the aerosol layer. On the other hand, the algorithm is unstable when an aerosol layer is not optically thick enough to screen the surface signals or when multiple aerosol layer exists. Analyses of circumstances whether the stereoscopic ATH algorithm works well or not are discussed in Section 4.2.

## 3.2 Resampling

For a stereoscopic ATH algorithm, two different instruments at different locations act as the two eyes of a human observer. Their different perspectives make one image from a satellite distorted relative to the other. In that case, it is difficult for computers to recognize that the two satellites are seeing the same object. Therefore, the geolocation of one satellite must be fixed relative to the other. Here, the image to be resampled is referred to as "A", and that of the reference geolocation as "B".

Resampling makes use of a *k*-dimensional tree method to find the nearest points of A from the geolocation of B. The k-dimensional tree is a fast algorithm locating the nearest neighbors of a point (B) to a k-dimensional tree of points (A) (Maneewongvatana and Mount, 1999). After converting two images from the geographic (spherical) coordinate system to a Cartesian coordinate system, a k-dimensional tree based on image A is used to find the 10 nearest neighbors of image B. The resampled image of A is a simple average of the 10 nearest points of A within 5 km of point B.

## 3.3 Parallax estimation

Parallax is calculated using an image-matching of moving window correlation technique (Lee et al. 2010). The method finds matching aerosol features in two images by finding the best-correlated TOA reflectance image window from each satellite, assuming that an aerosol feature is optically thick enough to be distinguishable from the background surface. The window is therefore set around a pixel of AOD > 0.3. We set a fixed window of $33 \times 33$ pixels from image B, and correlation coefficients with same-sized moving windows in image A are calculated. A moving window moves from $-7$ pixels to $+7$ pixels in the latitudinal and longitudinal directions, starting from the same longitude/latitude index of the fixed window. Highly reflective clouds can interfere with correlation calculations. To minimize the effect of nearby clouds (or, embedded small clouds), pixels identified as cloud by AHI are removed from correlation coefficient calculation. Moreover, correlation is not calculated when cloud fraction of a moving window exceeds 20 %. Finally, the parallax of a lofted aerosol layer is defined as the distance between the centers of the fixed and moving windows with the highest correlation coefficient among 225 moving windows. If the highest correlation coefficient does not exceed 0.9, the pixel is excluded from the retrieval.

Window size may influence the correlation between the two windows. A window size too small would not be optimal in deciding whether the images are the same; with too large a window, radiance from an adjacent cloud or a distinct land feature causes respectively higher or lower correlation coefficients. Meanwhile, the moving range of the windows dictates the maximum retrieved aerosol height. The moving range of $\pm 7$ pixels in both the latitudinal and longitudinal directions means that maximum parallax would be the distance equivalent to $7\sqrt{2}$ pixels. In this study, the size and moving range of the windows were decided empirically.

Because the method calculates distances between each grid point, the calculated values of parallax are discrete. The distance of a window center from the nearest window center is ~1 km, and the next nearest is at ~1.414 ($\sqrt{2}$) km near the satellite nadir point. The closest value of parallax is thus discrete at ~0.4 km difference. This discrete parallax is converted to height by a parallax–height conversion relationship. Gaps in parallax values are larger with increasing distance from the nadir point, so retrieved heights are not necessarily spatially continuous.

## 3.4 Parallax–height conversion

A graphical description of the parallax–height conversion process is given in Fig. 2. GEO orbits are always above the Equator, so the 2D conversion method used with LEO orbit satellites (e.g., MINX algorithm) had to be adjusted to a 3D

scheme. Assuming an aerosol layer at point P, for a layer of height of $h$ above a ground point P', the parallax is the distance between the ground scanning points of satellites A and B (A' and B'). Based on the longitude and latitude of A' and B', the zenith and azimuth angles at each point are calculated. The determination of aerosol height can then be summarized as follows:

$$(\overline{A'P'}) = h \tan \beta , (\overline{B'P'}) = h \tan \alpha, \tag{1}$$

$$\angle A'P'B' = \gamma - \theta. \tag{2}$$

From the cosine law:

$$d^2 = h^2(\tan^2 \alpha + \tan^2 \beta) - 2h^2 \tan \alpha \tan \beta \cos (\gamma - \theta), \tag{3}$$

and

$$h = \frac{d}{\sqrt{\tan^2 \alpha + \tan^2 \beta - 2 \tan \alpha \tan \beta \cos(\gamma - \theta)}}, \tag{4}$$

where $h$ is ATH, $d$ is the parallax, $\alpha$ and $\beta$ are the viewing zenith angles of A and B, and $\gamma$ and $\theta$ are the viewing azimuth angles of A and B. Geometrical values ($\alpha, \beta, \gamma$, and $\theta$) are determined by the locations of satellites A and B and the ground position of the aerosol layer. Based on the estimated parallax (Sections 3.2 and 3.3), the top height of the aerosol is then retrieved.

## 4 Sensitivity and uncertainty analyses

Retrieval sensitivity and uncertainty assessments used a different method from spectroscopic aerosol height retrievals which needs expensive radiative transfer models to simulate observations. For example, a sensitivity study of aerosol layer height (not ATH) retrieval from the $O_2$ A band (Hollstein and Fischer, 2014) included spectral resolution, instrumental noise, and surface inhomogeneity. Lee et al. (2015) assessed the retrieval uncertainty by giving perturbations on possible error sources such as aerosol optical properties and surface elevations. Meanwhile, the sensitivity of the stereoscopic ATH retrieval algorithm was based on the proficiency of parallax calculations, which are a function of aerosol height and the distance between two satellites. Moreover, parallax calculations only use the spatial patterns of observed radiances from satellite images. This implies an advantage in using the parallax of two satellite images, in that the geometrical method does not suffer from sensor calibration problems, with the unstable radiometric performance of AGRI (Zhong et al., 2021) being unlikely to affect the results.

Quantitative sensitivity and uncertainty assessment of stereoscopic ATH retrieval is done with a simple geometrical estimation of parallax. Since false registration of a satellite grid introduces error in parallax calculation, uncertainty simulating satellite INR error needs to be calculated. The uncertainty from the INR error of an instrument is considered by shifting the reference image (AHI) by 1 km. Intrusion of the INR shift during the retrieval process falsely locates level 1B

images and affects parallax calculations, which are directly related to an error in ATH retrieval. In addition, qualitative assessment of retrieval uncertainty by simulating a retrieval with AHI and AGRI is shown by simulating various retrieval situations such as bright underlying surface and multi-layer aerosols.

## 4.1 Sensitivity

Parallax is greater when the aerosol layer is at a high altitude or when the two satellites are farther apart (Fig. 2). The theoretical parallax variation with aerosol height and satellite location can be calculated for quantitative assessment of retrieval sensitivity using Eq. (3). Based on the viewing geometry of Seoul, South Korea (37° N, 127° E), and setting one satellite imager as the AHI (140.7° E), the calculated parallax according to the location of the other satellite and the altitude of the aerosol layer is as shown in Fig. 3. Using two satellite images with 1 km spatial resolution, parallax distances of <1 km cannot be resolved (Section 3.2). Considering that the image resolution coarsens with increasing distance from the sub-satellite point, the possible minimum parallax that can be resolved from two satellite images will be >>1 km. It follows that a pair of 1 km image-resolution satellite imagers are unable to retrieve aerosol heights that demand resolution of a parallax of <1 km. As the location of the other satellite imager approaches AHI, the parallax decreases rapidly (Fig. 3). This implies that a set of satellites too close to each other involves less parallax, which is a challenging condition for geometrical height retrieval. For example, an aerosol layer with a height of 2 km produces ~2 km of parallax using an AHI–AGRI pair at 104.7° E, which is sufficient for 1 km resolution. However, an AHI–AMI pair at 128.2° E cannot resolve this aerosol layer, as the parallax is ~0.75 km. To retrieve a lower aerosol height, satellites must be farther apart. When an aerosol layer is at lower altitudes, the parallax gradient is greater when the two satellites are closer, which means that the closer the pair of satellites, the greater possibility of small uncertainties during parallax calculations may produce larger errors in retrieval height. The parallax gradient becomes linear as the aerosol layer height increases. Two satellite imagers with insufficient separation are thus unfavorable for geometrical height retrieval. However, as the satellites get farther apart, the spatial resolution becomes coarser within their overlapping area. This implies that resampling image distortion with respect to reference image is greater, so that the possible minimum parallax may be greater than expected and the minimum height that a pair of satellites can resolve would be greater in the off-nadir region. Additionally, regions with large viewing angles have the same problem of coarsening spatial resolution. Therefore, we fixed the study region to East Asia, where dust transports and urban aerosol pollution are observed having proper viewing angles for each instrument.

Viewing geometries ($\alpha$, $\beta$, $\gamma$, and $\theta$; Fig. 2) are also functions of location on Earth. A contour of theoretical minimum height that two pairs of satellites (AHI–AGRI and AHI–AMI) can resolve using 1 km spatial-resolution bands is shown in Fig. 4. The theoretical minimum retrievable height decreases with distance from the center point of the two satellites. In terms of latitude, viewing zenith angles ($\alpha$ and $\beta$ of Eq. (4)) increase with distance from the Equator. For longitude, the difference in viewing azimuth angles ($\gamma - \theta$ in Eq. (4)) is lowest at the central longitude of the two satellites. For the AHI and AGRI (Fig. 4a), the minimum height differs by a few hundred meters in East Asia, so the location on Earth would have little impact on

the results. However, the AHI and AMI are too close together to retrieve aerosol heights of <3 km over the Yellow Sea. Furthermore, the minimum retrievable height gradient along location is larger for the AHI–AMI pair, which means that retrieval error caused by parallax uncertainty is more complex for this pair of satellites. Therefore, the AHI–AGRI seems to be a better choice for stereoscopic aerosol height retrievals over East Asia. Results for the two satellite pairs (AHI–AMI and AHI–AGRI) are further compared in Section 5.

**4.2 Uncertainty analysis**

    ATH retrieval uncertainties induced by 1 km latitudinal and longitudinal INR shifts are shown in Fig. 5. Uncertainty is defined as the difference between original ATH and height calculated using parallax considering 1 km shifts in AHI geolocation. The magnitude of uncertainty is larger when INR error is present in the longitudinal direction. When retrieving ATH for an aerosol layer reaching up to 5 km altitude, the possible retrieval error caused by 1 km longitudinal INR error is

~80 m, or about four times the maximum retrieval error for 1 km latitudinal INR error. The effects of latitudinal and longitudinal INR error on ATH retrieval uncertainty are thus quite different. In the latitudinal direction, the retrieval height uncertainty increases with the distance between the two satellites, whereas longitudinal INR error causes equal retrieval uncertainty regardless of the separation of the two satellites. This difference arises from the use of geostationary satellites. A longitudinal INR shift can simply be regarded as one satellite moving toward or away from the other along the Equator.

Although the height retrieval algorithm is more robust regarding parallax calculation errors when the two satellites are far apart (Section 4.1), the retrieval error itself increases with the distance between the two satellites when latitudinal INR errors are present. However, considering the actual INR errors of the satellites (approximately 0.5, 1, and 4 km at channels with 1 km resolution for AHI, AMI, and AGRI, respectively), the INR error would not be of concern for the retrieval of aerosol heights of a few kilometers.

Aside from the potential systematic error due to INR error, we also conducted an assessment of specific circumstances that may impact the retrieval of stereoscopic ATH. Fig. 6 illustrates a case study of retrieval simulating a reference image (AHI) and a moving window (AGRI; remapped onto AHI coordinates). We assumed 2-dimensional Gaussian aerosol layers at a height of 2.28 km with a peak albedo of 0.3 (Fig. 6b) and 0.1 (Fig. 6c) above a surface with average albedo of 0.06 (Fig. 6a). The reference image with an aerosol layer of 2.28 km over a dark surface shows the best correlation with a moving window

at +1 pixel to the longitudinal and +2 pixels to the latitudinal direction from the center of the reference image. However, in real situation, aerosol layers can have complex horizontal features that do not follow 2-dimensional Gaussian shapes, which can complicate the retrieval process. So, the numbers presented in this analysis are merely examples, because quantifying retrieval uncertainty for each circumstance is complicated. In Fig. 6b, a thick aerosol layer blocked the surface albedo pattern, so the retrieved height was 2.28 km, which is a desirable result. However, the surface signal penetrates a thin aerosol layer in

Fig. 6c, so that the moving window is matched at the center of the reference image, resulting in the retrieved ATH of 0 km. The assessment of retrieval results of multiple layers of aerosol is more complex. In those cases, the result depends on the

density of an aerosol layer above the other layer. If the upper layer is not dense enough to totally block the underlying features, then the result of the algorithm gives the height closer to the lower layer. Fig. 7 helps understanding the retrieval of ATH of multiple layer aerosol. Fig. 7a shows a single layer aerosol at 2.28 km over a dark surface (albedo = 0), and the retrieved height result is 2.28 km as expected. To simulate multi-layer aerosol ATH retrieval, aerosol layers at 4.57 km (the theoretical best correlation is at longitudinal +2 pixels and latitudinal +4 pixels from the center of the reference image) with different albedos are added. In Fig. 7b, a thin upper layer does not change the retrieved height. On the other hand, an opaque aerosol layer alters the best matching location on the reference image closer to the upper layer (Fig 7c and d). Considering that the retrieval result is the height of the upper layer, multiple layers results in underestimation of the ATH.

## 5 Results and Discussion

### 5.1 Comparison with CALIOP aerosol profile

For qualitative comparison of retrieved ATH with CALIOP, every algorithm requires a proper definition of aerosol height from CALIOP profile data. Definitions of aerosol height using the same lidar data differ according to how each algorithm defines its aerosol height products. For example, Lee et al. (2015) developed Aerosol Single scattering albedo and layer Height Estimation (ASHE) algorithm and compared their retrieval results with ATH based on CALIOP aerosol total backscattering coefficient (TBC) profile, with ATH being the altitude with a TBC of 0.03 $km^{-1}$ $sr^{-1}$ from the top of the profile. Lee et al. (2021) then developed a new, near production-ready ASHE algorithm, that no more retrieves ATH because of detail differences from original ASHE algorithm. So, they had to compare the results with newly defined CALIOP aerosol height, which is an aerosol extinction-weighted mean height. Considering the image-matching process shown in Section 3.3, results from the stereoscopic algorithm need to be defined at altitudes where horizontal texture is found. However, assessment of horizontal texture is challenging because CALIOP profile data provides 1-dimensional information along the horizontal plane with small footprint of ~70m. We instead generated a CALIOP aerosol height that is close to the top of CALIOP aerosol profiles because with high aerosol loading, which is a favorable condition for the retrieval, the algorithm is supposed to give a height near the top of an aerosol plume (c.f. Section 3.1). The height where a cumulative total extinction coefficient in the 532 nm channel (EC532) shows 90 % of the total (CALIOP 90 % extinction height) is compared with the stereoscopic ATH.

### 5.1.1 Case studies

Fig. 8 shows a comparison of stereoscopic ATH products from the two pairs of geostationary imagers, alongside a CALIOP aerosol extinction profile for 23 January 2020 as CALIOP passed over the western Yellow Sea (Fig. 8a). The AHI AOD was retrieved over Beijing, Shandong Peninsula, and the ocean between the Beijing and Shandong peninsulas. The stereoscopic ATH retrieval was retrieved mainly over the ocean, with AOD > 0.6 (Fig. 8b). Results for the two different pairs of

geostationary imagers (AHI–AGRI and AHI–AMI) are shown in Fig. 8c and d. The ATH retrieved from the AHI–AGRI pair was ~2 km, and that from AHI–AMI was ~4 km. ATH retrieval along the CALIOP path is shown with CALIOP EC and height in Fig. 8e. To collocate the ATH product with CALIOP geolocation, stereoscopic ATHs within 5 km of CALIOP

ground pixels were averaged. The ATH from the AHI–AGRI pair agreed well with the CALIOP 90 % extinction height of 1–2 km on collocated pixels. The ATH retrieved from the AHI–AMI pair near 37.5° N seems to have similar values to CALIOP (or the ATH of AHI–AGRI). The scatterplot (Fig. 8f) shows CALIOP 90 % extinction height versus stereoscopic ATH. The percent within expected error (EE%) (1 km) represents the fraction of stereoscopic ATH within 1 km of CALIOP 90 % extinction height in total collocated pixels. In this case, all valid AHI–AGRI ATH retrievals were within 1 km of

CALIOP 90 % extinction height, whereas for AHI–AMI ATH, only 18.5 % of the total collocated ATHs were within 1 km of CALIOP 90 % extinction height, and 33.3 % were within 1.5 km. Furthermore, several AHI–AMI ATH points have values of zero because of a lack of sensitivity to parallax induced by aerosol layers at <2 km height. Considering that the collocation of stereoscopic ATH and CALIOP 90 % extinction height involved the averaging of ATH values, values out of retrieval sensitivity (<3 km) may be products of averaging values of 0 km and 4 km near the CALIOP path.

The case of 8 April 2020 is shown in Fig. 9. A thick aerosol layer with AOD > 1.0 (reaching near 2.0 at the thickest part of the layer) was under a cloudy sky in Jinan, China, with low-level thin clouds along 32° N latitude (Fig. 9a, b). Unlike clouds over the northern part of the area, ATH overestimation by these thin clouds does not appear high enough to be screened out during the quality control procedure for the AHI–AGRI pair. Results (Fig. 9c) indicate that the retrieved ATH over the area spans from 2 to 5 km. The ATH results for the AHI–AMI pair (Fig. 9d) are similar. The actual profile of aerosol ECs

observed by CALIOP (Fig. 9e) suggest that the retrieved ATH for both pairs spanning from 2 to 5 km appears reasonable. Results for the AHI–AGRI pair agree more with CALIOP 90 % extinction height at most collocated points. The EE% values for 1 km within CALIOP 90 % extinction height for AHI–AGRI and AHI–AMI are 72.5 % and 61.3 %, respectively (Fig. 9f); those for 1.5 km are 55.0 % and 32.5 %, respectively.

The sensitivity test results (Section 4) are consistent with actual retrieval cases. From these two cases of comparison between

ATH retrieval comparison and CALIOP, the performance of the stereoscopic ATH algorithm for an aerosol layer at lower or higher altitudes in the troposphere can be assessed. When an aerosol layer is under 3 km (as on 23 January 2020), the stereoscopic ATH algorithm with the pair of closer satellites (AHI–AMI) failed to retrieve aerosol height. However, if the satellite separation has sufficient sensitivity for the retrieval of aerosol heights of <3 km, the geometrical height retrieval algorithm for atmospheric structure is applicable. The AHI–AGRI ATH EE% on 8 April 2020 was significantly lower than

that on 23 January, possibly because the aerosol layer was thinner. Unlike the first case, the CALIOP EC profile of the latter case has few values of >0.3 cm$^{-1}$, indicating retrieval error caused by low aerosol loading. However, the possibility of systematic worsening of the retrieval accuracy of lofted aerosols remains because complex vertical features (e.g., multi-layer aerosols) make aerosol height retrieval challenging. Further data analysis is required to determine whether the deterioration in accuracy with higher ATH is systematic.

## 5.1.2 Long-term comparison with CALIOP

2-D histograms of CALIOP 90 % extinction height versus stereoscopic ATH (Fig. 10) shows the overall performance of the stereoscopic ATH retrieval algorithm. To assess robust results that are less affected by aerosol loading, ATH with a correlation between the reference image and the moving window (matching correlation) >0.95 are used for long-term analysis. Collocation was done in the same manner as the case studies. As expected, the general retrieval performance using the AHI–AGRI pair worked better than the AHI–AMI pair. RMSDs with CALIOP 90 % extinction heights are 1.66 km for AHI–AGRI, and 4.98 km for AHI–AMI. The fraction of ATH within 2 km from CALIOP 90 % extinction height was 88.9 % for AHI–AGRI and 57.4 % for AHI–AMI (for 1 km, 24.4 % and 5.9 %, respectively). During the study period, most cases in East Asia had CALIOP 90 % extinction heights of <3 km, which were beyond the sensitivity range of the AHI–AMI pair. The stereoscopic AHI–AMI ATH (Fig. 10b) has no values <3 km. The ATHs of the aerosol layer below the height of retrieval sensitivity were clustered around 4 km, with some at around 8 km. As shown in the sensitivity study, a small uncertainty in parallax estimation is followed by a great error in the height retrieval when using satellites close to each other. Better spatiotemporal matching of images that are taken closer in time with the finer resolution is needed from stereoscopic ATH retrieval with satellite pairs with less sensitivity.

Frequency distributions of the difference of stereoscopic ATH and CALIOP 90 % extinction height are shown in Fig. 11. The peak of the difference between AHI–AGRI ATH and CALIOP 90 % extinction height was near zero (Fig. 11a), with an average difference of -0.07 km, (standard deviation 1.66 km). The AHI–AMI ATH displayed peaks at 2, and 5 km, with an average difference of +2.56 km (standard deviation 3.60 km). The highest peak at 2 km was mainly due to overestimation of the lower aerosol layer. The increase in frequency at specific values was due to parallax estimation giving discrete values through image resolution. Although stereoscopic ATH values were spatially averaged during collocation with CALIOP, discontinuous spatial features may still occur. The difference between land and ocean results appears to be negligible for both pairs of satellites.

Fig. 12 shows the average difference between the CALIOP 90% extinction height and the stereoscopic ATH results according to the aerosol loading. The aerosol loading is calculated using the CALIOP extinction coefficient values integrated vertically in collocated pixels. Fig. 12a and b represent error analysis results for AHI–AGRI ATH and AHI–AMI ATH, respectively, while Fig. 12c and d show the error analysis results for stereoscopic ATH when the correlation between the reference image and the moving window is greater than 0.95 during the image–matching process. The overall root mean squared difference (RMSD) between AHI–AGRI ATH and CALIOP 90% extinction height in Fig. 12a was 2.64 km. As the aerosol loading increases, the RMSD decreases to 2.07 km. This is because, as mentioned in Section 4.2, surface feature signals become blocked as aerosol loading increases, leading to a more accurate stereoscopic ATH retrieval. The box-whisker plot also shows that the spread decreases as the retrieval becomes more stable, and the bias approaches 0 km. In Fig. 12c, selecting data with high matching correlation coefficients significantly

reduced the RMSD for cases with low aerosol loading. This indicates that when the underlying surface is dark or the vertical distribution of aerosol layers is simple, which is favorable for the stereoscopic ATH retrieval algorithm to work, the image-matching is robust. As a result, the overall RMSD reduced to 1.66 km. In Fig. 12b, the overall RMSD for AHI–AMI ATH was 4.74 km. As the aerosol loading increases, the RMSD decreases to 3.81 km and then increases again to 4.19 km. This seems to be because urban/industrial aerosols emitted at large cities in East Asia tend to reside at lower altitudes (Chang et al., 2023). Unlike the AHI–AGRI pair, the AHI–AMI pair lacks retrieval sensitivity when high aerosol loading resides at lower altitudes. In Fig. 12d, the overall RMSD is 4.42 km, and the RMSD values for some CALIOP integrated extinction bins are similar or even increased. This indicates that the lack of sensitivity in stereoscopic ATH using the AHI–AMI pair is a greater source of error than surface darkness or vertical distribution of aerosol. Therefore, selection of highly correlated matching data could not improve the retrieval quality. Fig. 13 shows the distribution of differences according to CALIOP 90% extinction height. AHI–AGRI ATH has a significant positive asymmetry for aerosol layers at height lower than 1 km, which appears to be a result of the retrieval sensitivity beyond the range of the AHI–AGRI pair. The AHI–AGRI ATH has good agreement with CALIOP 90% extinction height up to ~3 km, after which it shows a negative bias, and the box-whisker plot moves in a negative direction. This is because when the top of an aerosol layer is high up, its structure can be complex, or multiple layers of aerosol may exist, causing interference with signals from lower layer aerosols as described in Sect 3.1. The difference of AHI–AMI ATH with the CALIOP 90% height comes close to 0 (Fig. 13b, d). This is because the parallax which AHI–AMI pair can retrieve is formed at higher altitudes. Still, RMSD and the range of whisker is longer than for AHI–AGRI ATH. Meanwhile, using highly correlated matching result (Fig. 13d), the positive asymmetry of the box-whisker plots decreases, indicating that this data is more stable than data without considering matching correlation.

**5.2 Comparison with ground-based lidar**

The use of passive sensors onboard GEO satellites enables continuous monitoring of aerosol vertical features over a broad area. To assess the diurnal variation monitoring of hourly stereoscopic ATH retrieval, aerosol extinction profiles from KALION ground-based lidars are used. Examples of the stereoscopic ATH algorithm capturing hourly variations of aerosol vertical features are shown in Fig. 14. Comparisons with ground-based lidar data were undertaken using only the AHI–AGRI results, which proved to be most accurate. For the spatial collocation, valid retrievals within 5 km of ground-based lidar sites were averaged. For 7 April 2020 (Fig. 14a and b), ATH was compared with observations at the SNU station. The hourly ATH map (Fig. 14a) indicates the initiation of transport of an aerosol layer with the ATH of >7 km. Upon reaching the Korean Peninsula, the aerosol layer descended and dissipated. According to aerosol extinction data from ground-based lidar,

the aerosol layer became thinner both geometrically and optically. The hourly variation of AHI–AGRI ATH began from 2.3 km at 01:00 UTC, reaching 1.7 km at 07:00 UTC. The ATH plunged to 1.1 km at 02:00 UTC, and the stereoscopic ATH retrieval algorithm captured its descent. The ATH maps (Fig. 14a) seem continuous, so the inconsistent value at 02:00 UTC may have been caused by a sampling problem. Another possibility is the error caused by the difference of observation time between the two sensors. The AHI performs a full-disk scan in 10 minutes, whereas the AGRI takes 15 min, starting at the top of the hour. Therefore, when strong horizontal air motion is present, a small difference in observation time between two imagers may have a significant impact on parallax calculations during stereoscopic ATH retrieval. For instance, aerosol long-range transports involve strong westerlies in East Asia, so the observation time difference may also be the source of retrieval error. Lee et al. (2018) addressed this problem by interpolating observation times from one satellite to match the other. Considering that the percentage of stereoscopic ATH within 1 km from CALIOP 90 % extinction height was 46.2 %, the fluctuation at 02:00 UTC may simply be regarded as the expected retrieval error. On 9 April 2020 (Fig. 14c and d), ATH was compared with aerosol extinction at Gosan station (GSN). In that case, ground-based lidar profile shows a multiple aerosol layer with one stable layer at ~1 km and the other at 2-3 km. Except for 04 UTC, the stereoscopic ATH reasonably follows altitudes of the upper layer. At 04 UTC, the extinction coefficient of the upper layer decreases, then ATH shows large error as described in Section 4.2. Use of geostationary satellites enables the analysis of diurnal variations in ATH. The former case indicates how the vertical structure of an aerosol layer changes during long-range transport, whereas the latter represents a case of a multiple aerosol layer.

**6 Conclusions**

A stereoscopic ATH retrieval algorithm was developed on the basis of a geometrical height calculation method applying to any structure in the atmosphere. The advantages are that this type of method does not depend on sensor calibration and it does not require expensive radiative transfer computations. Furthermore, the method is not affected by variations in aerosol optical properties when the image-matching method works well. As a result of a sensitivity study, a well-separated satellite pair was found to be a better choice for stereoscopic ATH retrieval. Coarser spatial resolution in off-nadir areas can cause reductions in retrieval sensitivity. Retrieval uncertainty due to longitudinal INR shifts is greater when the two satellites are further apart, whereas retrieval uncertainty due to latitudinal INR shift is a function of aerosol height alone. An INR performance shift of 1 km introduces a retrieval error of a few dozen of meters, which is negligible when retrieving ATH of a few kilometers. Two case studies showed that the retrieval results of an AHI–AGRI satellite pair were consistent with the CALIOP EC profile. The general performance of the stereoscopic ATH retrieval was also better with the AHI–AGRI pair, with EE% values of 88.9 % and 57.4 % for 2 and 1 km, respectively. For the AHI–AMI pair, EE% values were 24.4 % and 5.9 % for 2 and 1 km, respectively. The mean bias in ATH from the AHI–AGRI pair was -0.07 km, whereas the AHI–AMI pair showed a strong positive bias of +2.56 km on average, with a peak at +2 km due to a lack of sensitivity at lower aerosol layers. Error analyses revealed that high aerosol loading facilitates image-matching, thereby reducing RMSD from CALIOP

90% extinction height. However, dark surface and simple vertical distribution of aerosol layer lead to successful image-matching in low aerosol loading situation. In the case of AHI–AGRI, the lack of retrieval sensitivity was larger source of difference than aerosol loading. Meanwhile, aerosol layers reaching higher altitudes with complex features, such as multiple layers, were another source errors, for which the RMSD increased at high 90% extinction height. Comparison with ground-based lidar revealed an ability to monitor diurnal variations in aerosol height with the synergetic use of geostationary satellites. Analysis of the sensitivity and uncertainty of the stereoscopic algorithm and its application to three geostationary satellite images over East Asia confirmed the capability of ATH retrieval using geometrical parallax calculations. Future work will include additional lower-level cloud screening, consideration of the difference in pixel-level scan times between two satellites, and more complex parallax–height conversion using spherical trigonometry. We expect the use of 500 m resolution red band observations with the stereoscopic ATH retrieval algorithm to provide greater sensitivity for lower-level aerosol layers with improved accuracy.

*Code availability.* The aerosol top height product data from AHI, AMI and AGRI are available on request from the corresponding author, Jhoon Kim (jkim2@yonsei.ac.kr).

*Author contribution.* MK, HL, and JK designed the experiment. MK carried out the data processing. HL, SL and YC provided support for data. MK wrote the manuscript, with contributions from all co-authors. JK reviewed and edited the article. JK provided support and supervision. All authors analyzed the measurement data and prepared the article.

*Competing interests.* The authors declare that they have no conflict of interest.

*Acknowledgements.* We thank all principal investigators and their staff for establishing and maintaining the KALION sites and CALIOP used in this investigation. The authors acknowledge the Chinese National Satellite Meteorological Center for the satellite data. This work was supported by a grant from the National Institute of Environment Research (NIER), funded by the Ministry of Environment (MOE) of the Republic of Korea (NIER-2022-04-02-088). Authors also would like to acknowledge the support by Samsung Advanced Institute of Technology (SAIT).

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

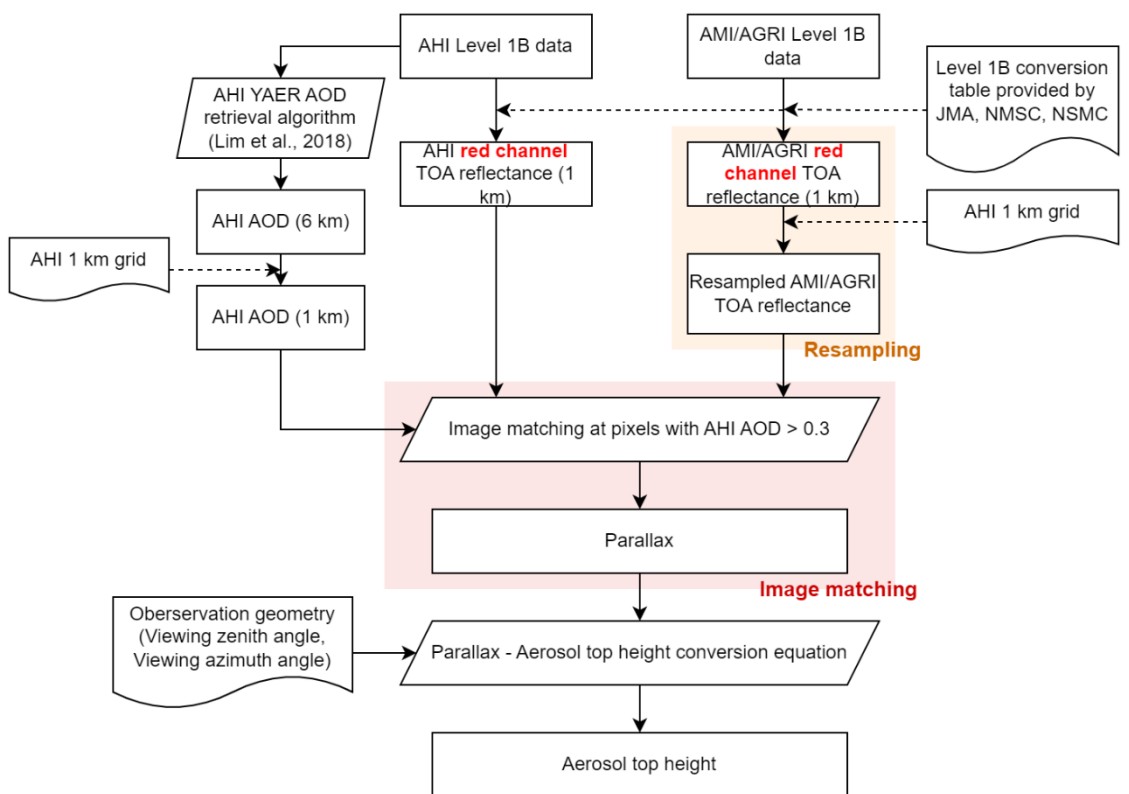

**Figure 1: A flowchart of the stereoscopic aerosol top height (ATH) retrieval algorithm.**

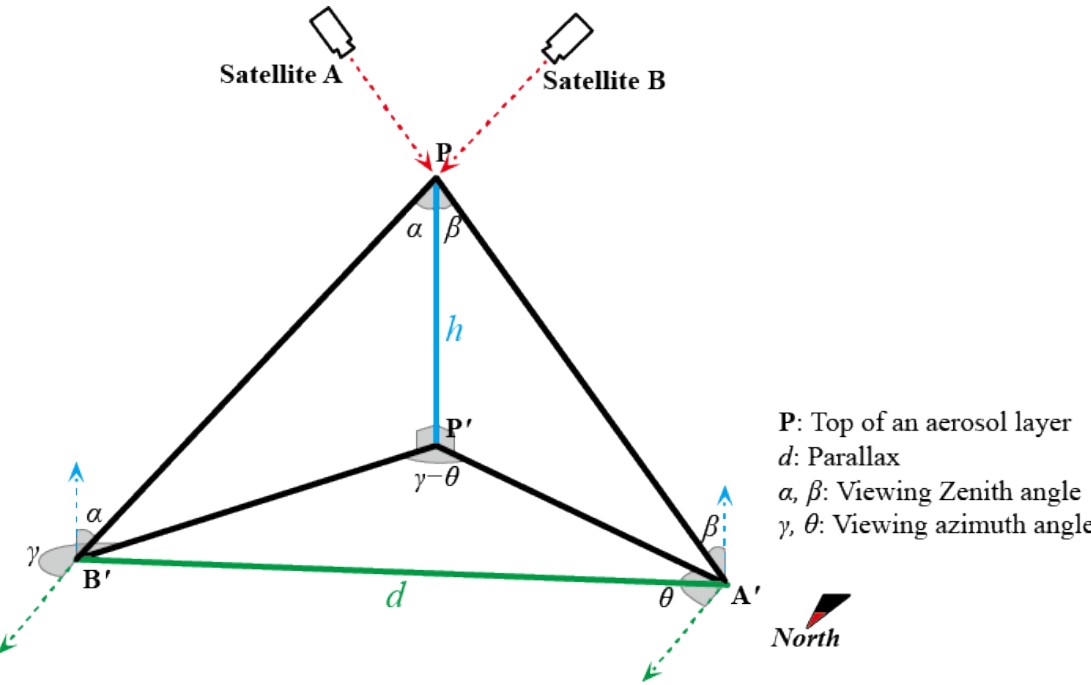

**Figure 2: Graphical depiction of parallax from the top of the aerosol layer (P) observed using a pair of satellites (A and B).**

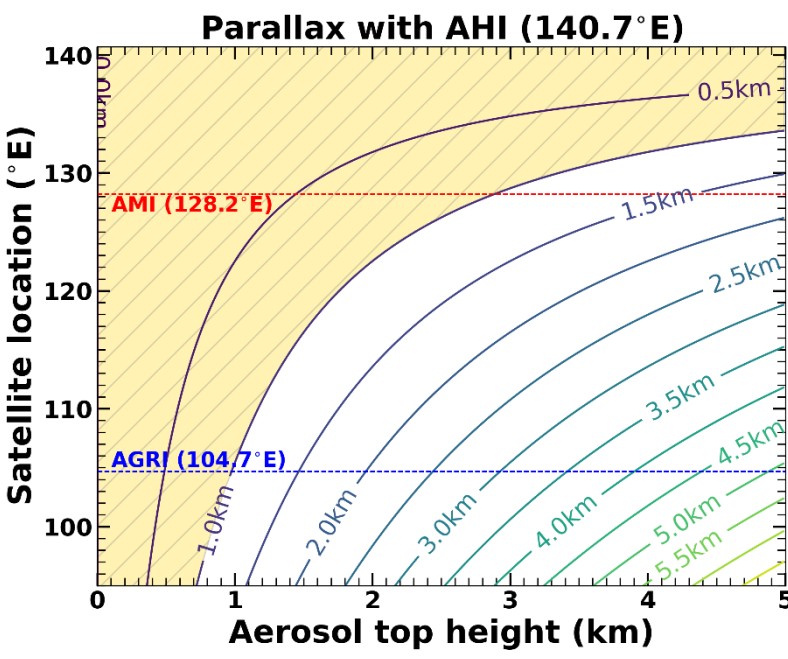


**Figure 3: Contour of theoretical parallax when one satellite in a pair is AHI and the other moves from 95° E to 140° E, according to the height of target aerosol layer. Yellow shading indicates parallax values that cannot be resolved using 1 km resolution imagers.**

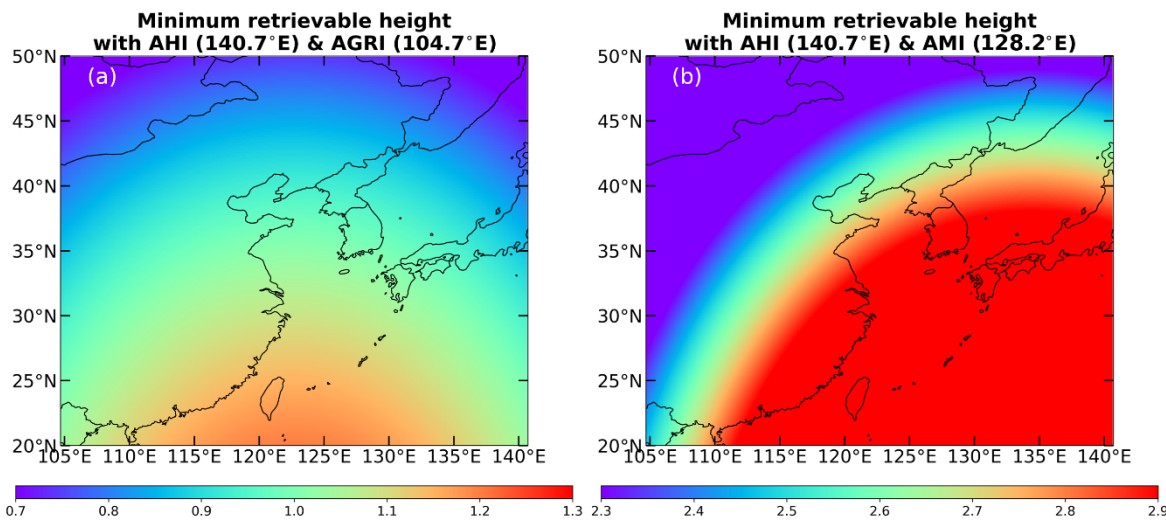

Figure 4: Minimum retrievable aerosol height from the (a) AHI–AGRI and (b) AHI–AMI (b) satellite pairs. The minimum retrievable height is set for 1 km resolution images to resolve parallax.

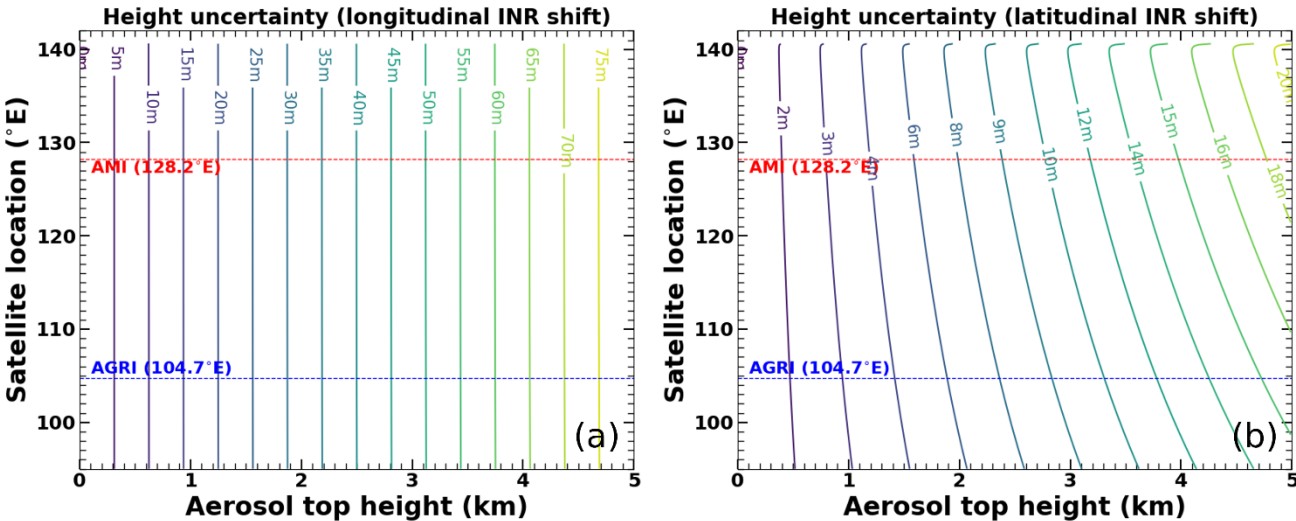

**Figure 5: Height retrieval uncertainty assuming 1 km latitudinal (a) and longitudinal (b) shifts in AHI geolocation.**

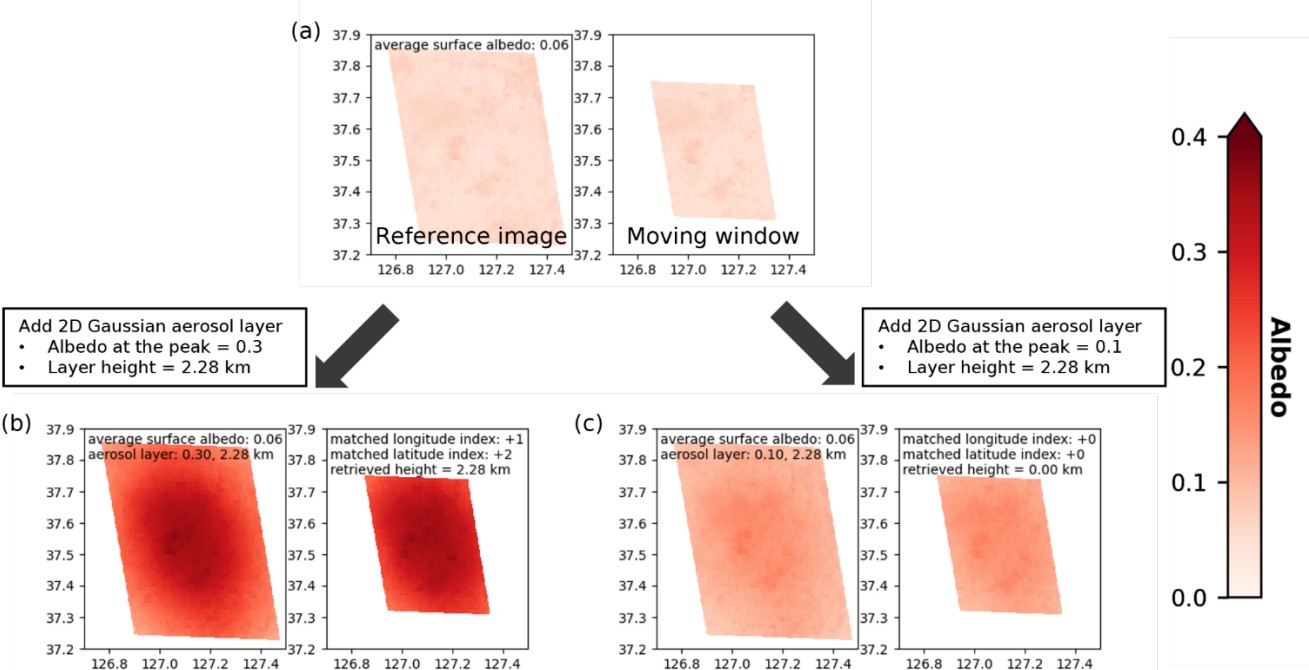

**Figure 6 A graphical illustration of stereoscopic ATH retrievals. Over (a) a surface with average albedo of 0.06, 2-dimensional Gaussian-shaped aerosol layers at 2.28 km with peak albedo of (b) 0.30 and (c) 0.10 are added.**

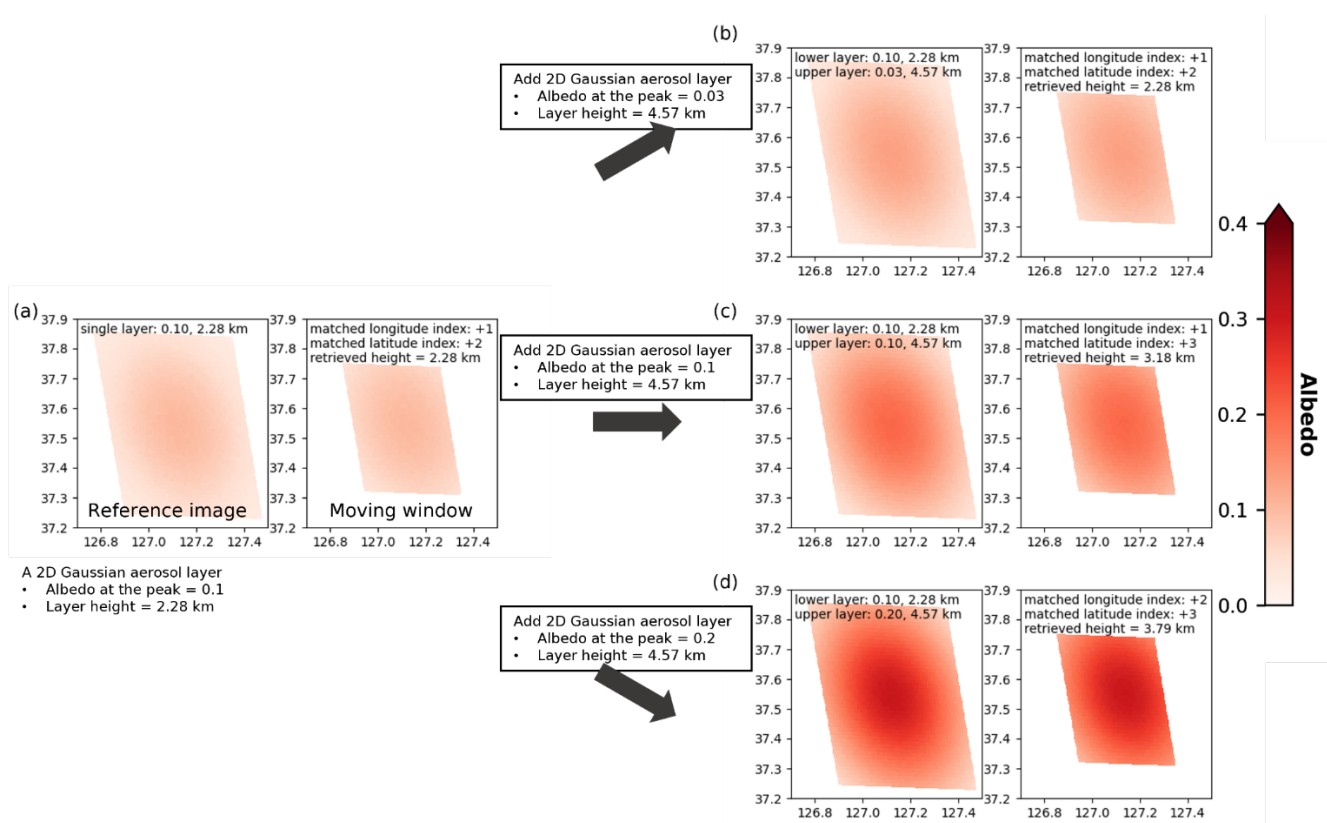

**Figure 7 Same as Fig. 6, but assuming a dark surface. Over (a) a single layer of aerosol with peak albedo of 0.10 at 2.28 km, three different upper aerosol layers at 4.57 km with peak albedo of (b) 0.03, (c) 0.10, and (d) 0.20 are added.**

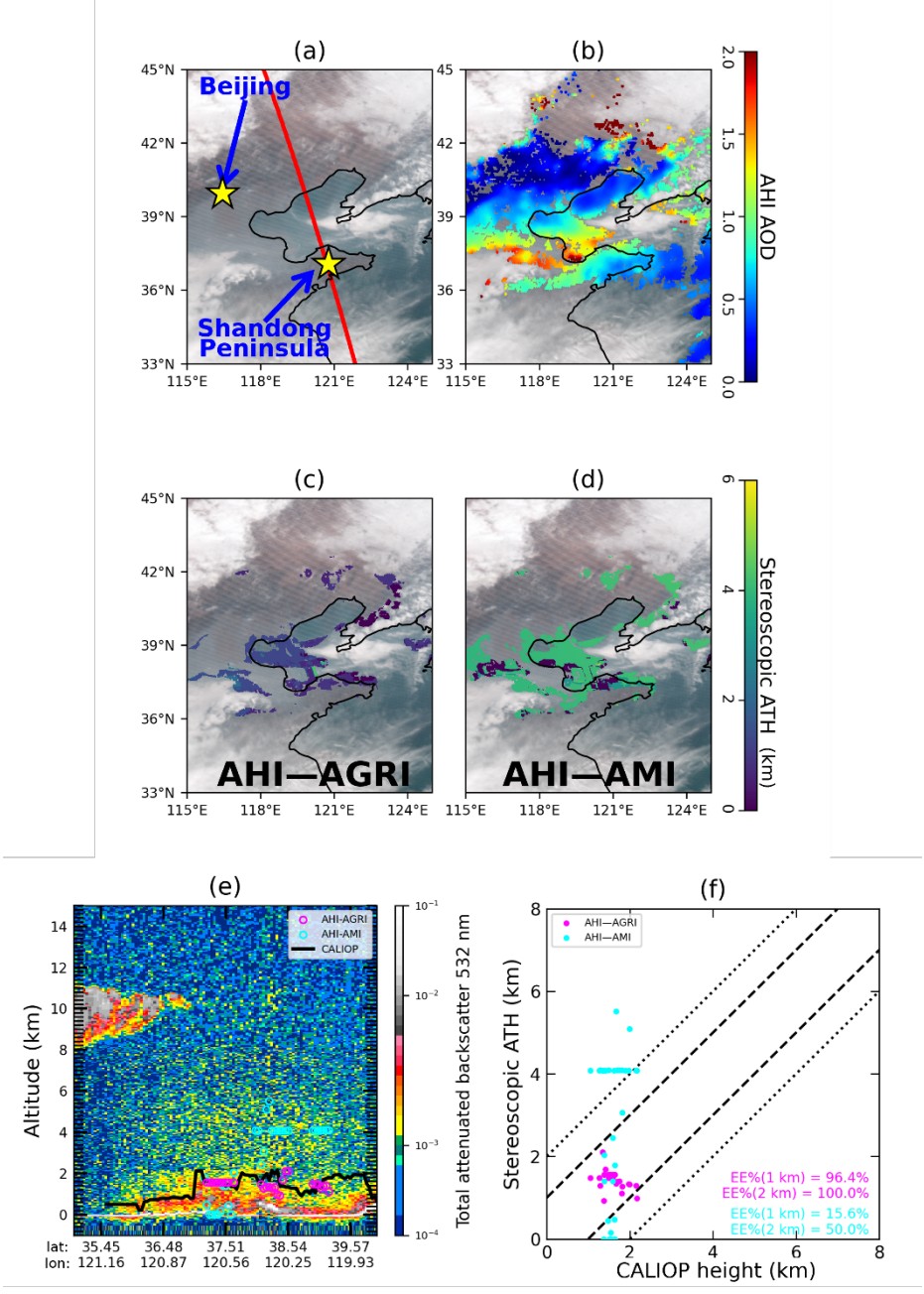


**Figure 8: A case of ATH retrieval on 23 January 2020. (a) RGB image of the area of interest (red line shows CALIOP overpass). (b) AHI AOD. (c) Stereoscopic ATH retrieved using AHI–AGRI. (d) Stereoscopic ATH retrieved using AHI–AMI. (e) Stereoscopic ATH and CALIOP extinction coefficient profile (red dots represent ATH from AHI–AGRI, orange dots from AHI–AMI, and black line represents CALIOP 90 % extinction height). (f) Scatterplot of**
**CALIOP 90 % extinction height versus stereoscopic ATH.**

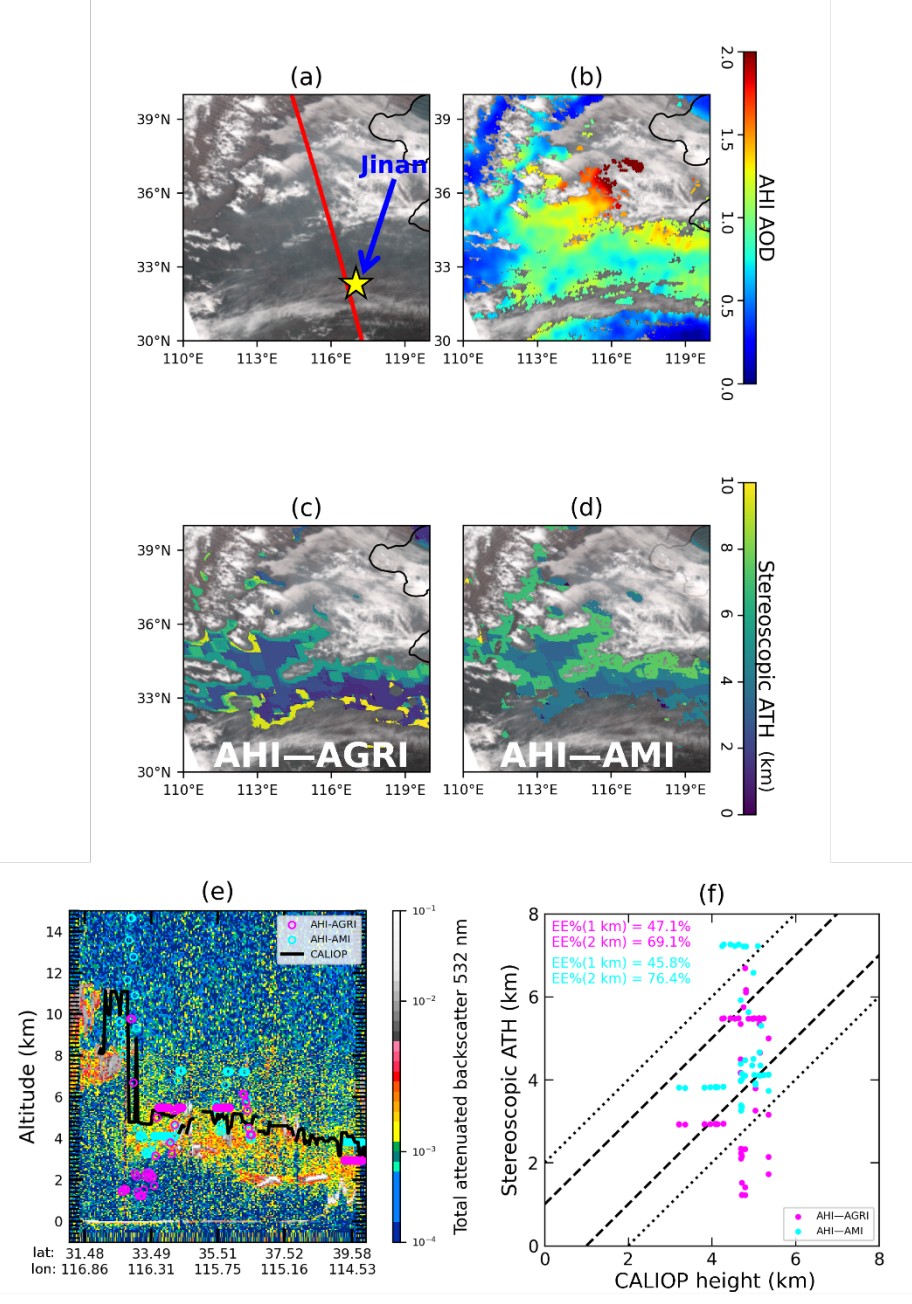

**Figure 9: As for Fig. 8, but for 8 April 2020.**

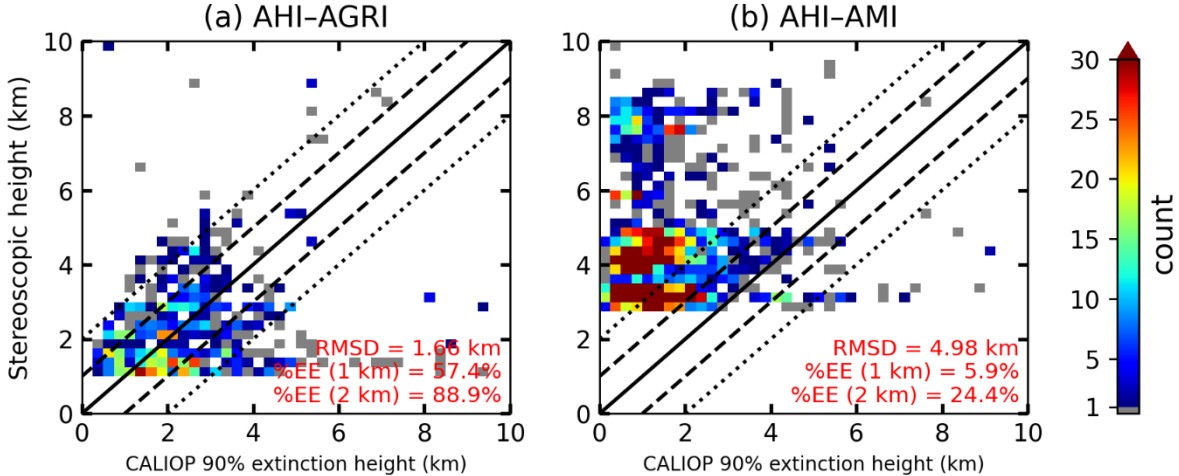

**Figure 10: 2-D histograms of CALIOP 90 % extinction height versus stereoscopic ATH of the (a) AHI–AGRI and (b) AHI–AMI satellite pairs.**

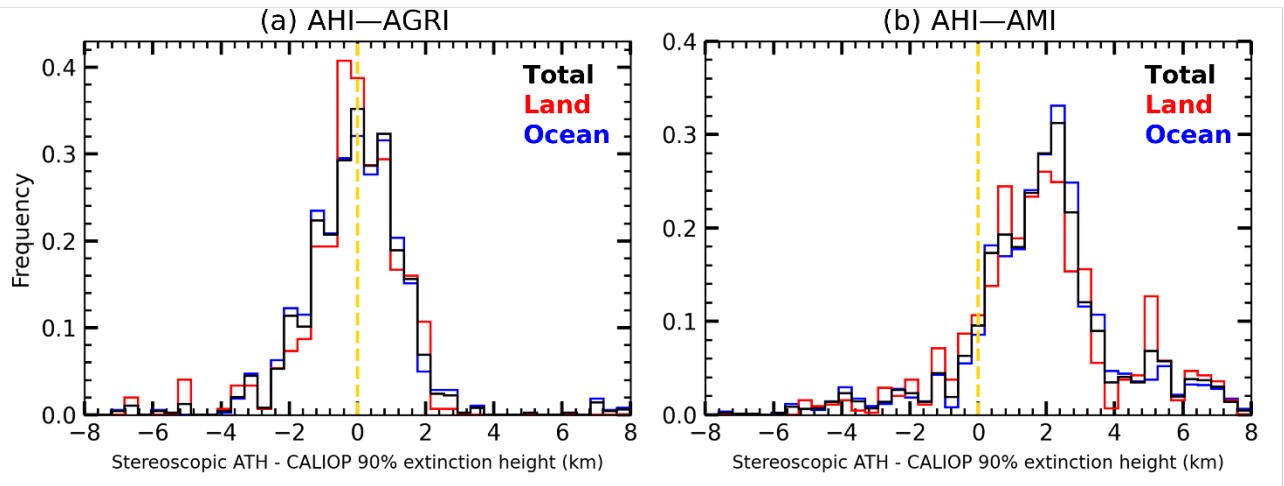

**Figure 11: Frequency distributions of the difference between stereoscopic ATH for the (a) AHI–AGRI and (b) AHI–AMI satellite pairs and CALIOP 90 % extinction height.**

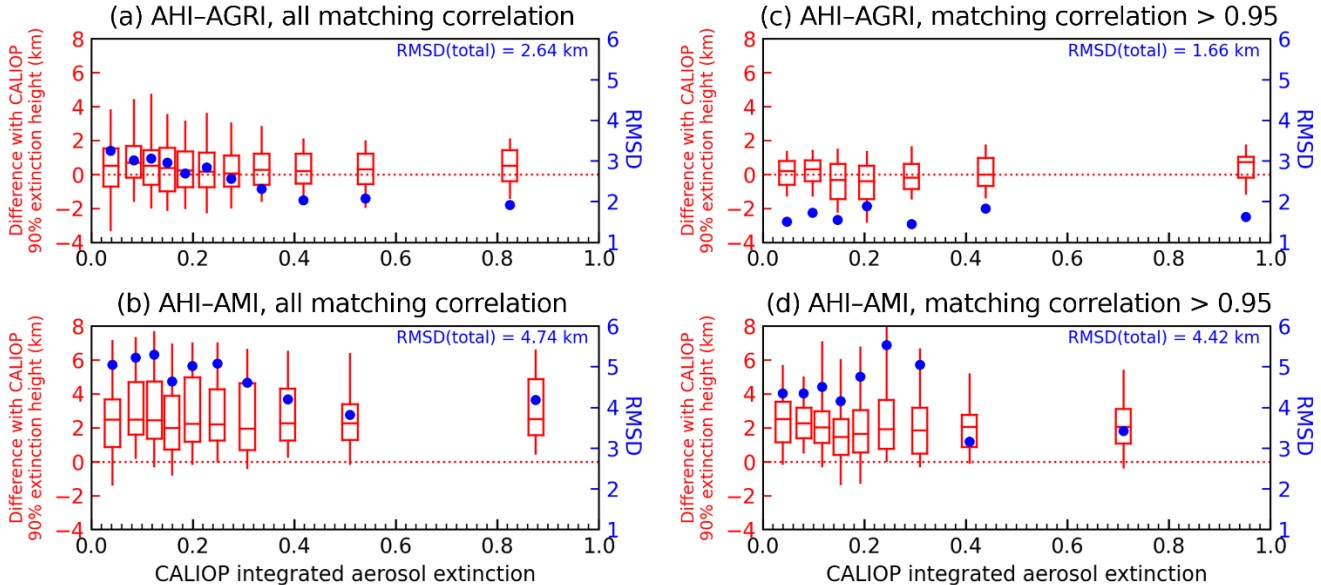

**Figure 12 Box-whisker plots of average difference and scatter plots of RMSD between stereoscopic ATH and CALIOP 90% extinction height according to CALIOP integrated aerosol extinction. Red box-whisker plots show 5th, 25th, 50th, 75th, 95th percentiles of (stereoscopic ATH – CALIOP height). Blue circles show RMSD of ATH and CALIOP height. All results are used for (a) AHI-AGRI ATH and (b) AHI-AMI ATH. Image-matching correlation coefficient of the best correlated moving window > 0.95 are used for (c) AHI-AGRI ATH and (d) AHI-AMI ATH.**

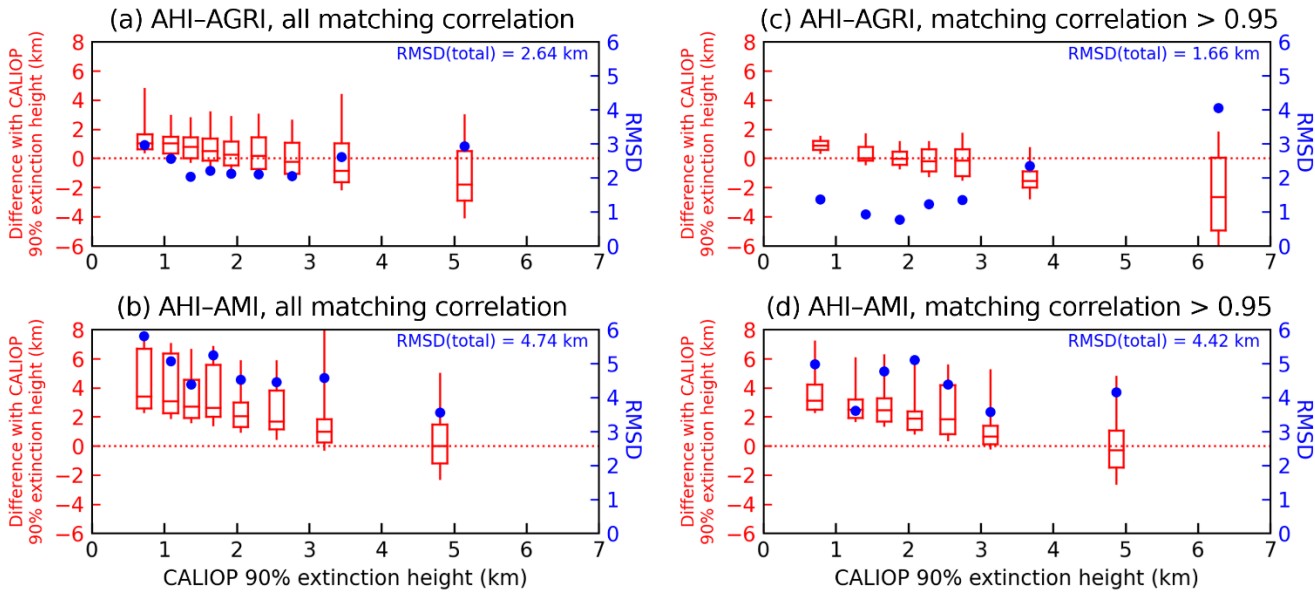

**Figure 13 Same as Fig. 12, but according to CALIOP 90% extinction heights.**

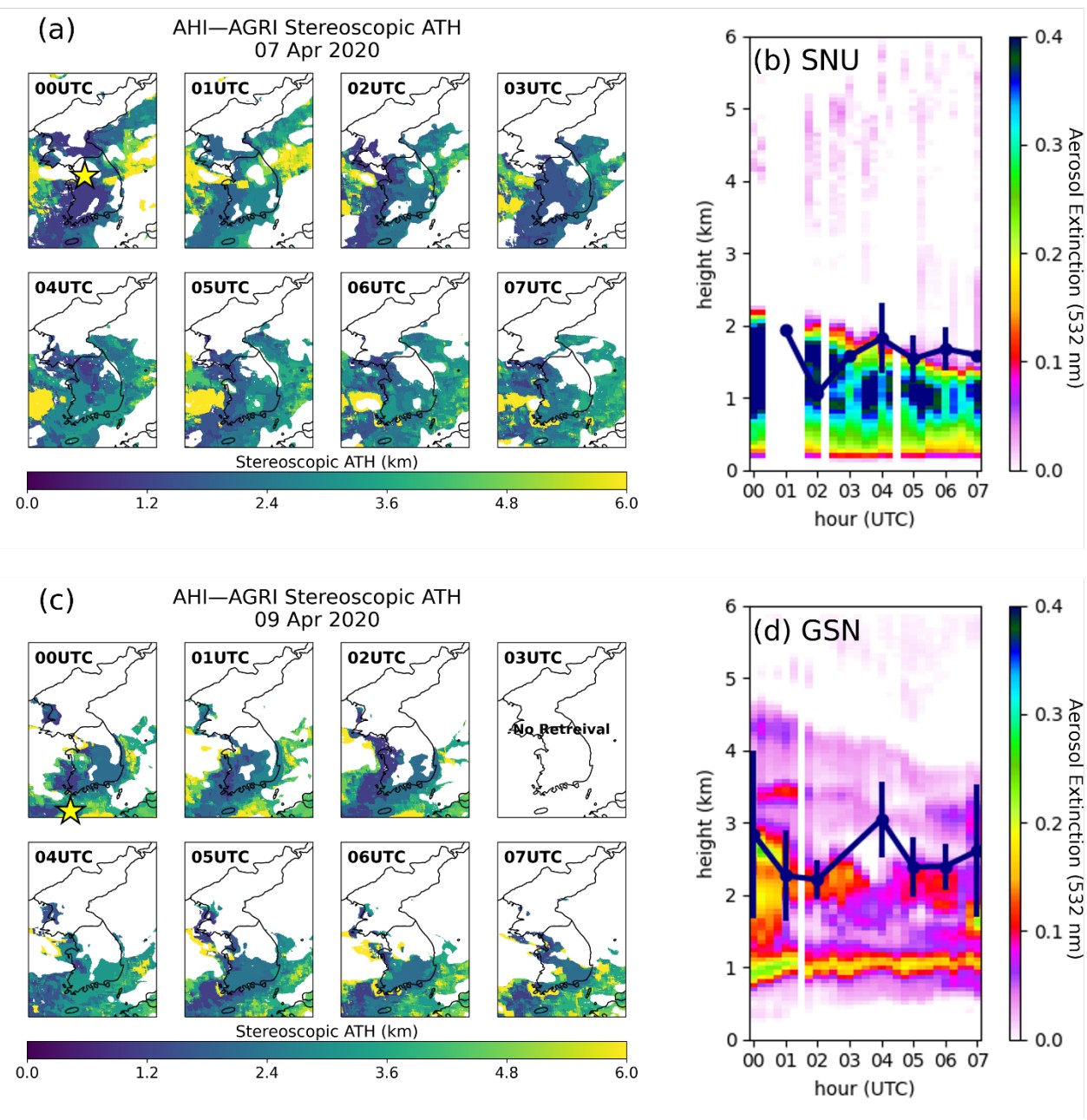

**Figure 14: (a) Hourly stereoscopic ATH maps for 7 April 2020. (b) Aerosol extinction profile observed at SNU on 7 April 2020. The navy line represents the ATH from AHI–AGRI. (c) As for (a), but for 9 April 2020. (d) As for (b), but for the GSN station. Yellow stars in (a) and (b) represents ground-based lidar stations at Seoul national university and Gosan, respectively.**