# Peer review of "Exploring geometrical stereoscopic aerosol top height retrieval from geostationary satellite imagery in East Asia"

_Atmospheric Measurement Techniques, 2022_

## Referee Comment (RC1)

Review for Atmospheric Measurement Techniques

Title: Exploring geometrical stereoscopic aerosol top height retrieval from geostationary satellite imagery in East Asia

By Minseik Kim, et al.

**General comments:**

This study assessed the application of different viewing geometries for a pair of geostationary imagers, AHI-AGRI and AHI-AMI to retrieve aerosol top height (ATH) information. The stereoscopic algorithm is presented, which converts the lofted aerosol layer parallax, calculated using image-matching of two visible images, to ATH.

What is strongly missing in the manuscript is a discussion on the required ATH quality for different applications. I am not an expert in utilizing the ATH data but knowing the PBL processes I assume that 1-2km offset between calculated and measured on the ground or retrieved from CALIOP ATH is too high and further improvements of the retrieval approach are needed to produce a product of the required quality.

The insufficient quality of a product is a cause for my decision to reconsider the manuscript after major revision, when a better (required for certain applications) quality of the product has been achieved.

**Specific comments:**

How is ATH defined in the study? How it differs from aerosol layer height? This question came to my mind on P8 L 232

P2, L 31. How narrow? Please, provide numbers or refer here to Sect. 2.2.1
P2, L 31. Please, specify more exactly bypasses time
P2, L 35. I suggest using the word "distribution" instead of "structure"
P2, L 59. Remove "data"
P2, L 62-64. Is it lack if channels or lack of the stereoscopic view, which is insufficient?
P4, Sect 2.1.2. Please, add bands characteristics, as in 2.1.1
P4, L 96. The Advanced Meteorological Imager (AMI) ….
P4, L 97. Please specify new channels if they are used in the study. If not, it is not necessarily to mentioned added channels here.
P4, L 99. Please, clarify: The AMI spectral bands are similar to those of AHI, except for a VIS and IR band; the center wavelengths and spatial resolutions of the VIS bands of AMI and AHI are similar.
P4, L 117-118. Please, rephrase
P4, L 120. CALIOP product is less accurate compared to the ground-based measurements. I suggest naming of the inter-comparison with CALIOP as evaluation, instead of validation. I also suggest discussing first an opportunity for validation with ground instruments and second mention the evaluation with satellites (which in general have an advantage in

coverage, though CALIOP coverage is quite small, but may allow evaluation in the conditions where ground instruments are missing)

P6, Sect. 3.1 Have you considered to develop two approaches, one for land and one for ocean, to resolve the ocean/land contribution at different wave lengths?

P6, L 160. Please, start with the definition of the parallax, then continue with the description of how is was calculated.

P6, L 167. Please, provide short definitions here

P7, L 191. Based on what the AOD lower limit of 0.6 was chosen?

P8, L 228. Based on what the limit of 10km for the ALH was chosen? Can all pixels in the moving window be checked on the presence of AOD data? This will allow avoiding the influence of clouds.

P9, L 242. What is INR error? Is it calculated based in instrument specifications?

P9, L 273. Agree. Why "a simple cause of retrieval uncertainty was involved here?" (L 242)

P10, L 302. Please add the definition for EC

P11, L 308. Please, replace "valid" with "retrieved" or "provided".

P11, L 309. Please, replace "valid" with "retrieved" or "calculated".

P11, L 306-309. Please add to the text the reference to Fig. 6a and 6b .

P11, L 310. The difference of 2 km in the ATH estimation is significant, when one think about the location of aerosol layer regarding the planetary boundary layer (PBL). The knowledge on that (within or above PBL) is important for predicting the further aerosol transport directions and intensity. This is more critical for high AOD loading episodes, which you consider.

Why 1 and 2km difference was chosen as criteria for evaluation? This is very big offset, if we think about possible applications of the calculated ATH. What are typical criteria for CALIOP ATH evaluation? Other ATH products?

P11, L 315-331. Can you discuss the conditions in which the disagreement between two products is most pronounced? And provide plot for AHI-AGRI vs AHI-AMI ATH.

P12, L 342 overestimated.... or aerosols were distributed evenly along the height

P13, Sect.5. To my understanding, collocation of geostationary satellite with ground measurements provides an opportunity for considerably higher number of collocations than with CALIPSO. However, only two cases are considered. To make a conclusion on the validation results, statistics (bias!) should be calculated using all possible collocations. Scatter plot as Fig.8 as well as frequency distribution plot are needed to be presented and discussed.

P13, L 397. Why AOD limit of 0.6 was applied, if "…. not affected by variations in aerosol….

Figure 4. Please, check the location of AMI
Figure 6 c,d. Please change the color scale to see better the difference between two pairs.

---

## Author Response (AR1)

We would like to thank the Reviewer for his/her thorough and detailed review as well as for the suggested papers. Our responses (in blue) for each comment (in black) are provided below.

**[updates in the algorithm]**

Lowered the retrieval cutoff AHI AOD from 0.6 to 0.3.

Removed 10 km ATH cutoff.

Image-matching is no more conducted when 20 % of the pixels in moving (or, reference) window is cloud-contaminated.

**[changes in figures]**

Figures use updated version of ATH.

CALIOP L2 aerosol extinction profile is replaced to CALIOP L1 total attenuated backscatter in Fig.6 (e) and Fig. 7 (e).

Fig. 8 is replaced with 2-dimensional histogram.

Added x=0 line in Fig. 9.

Changed colormaps in Fig. 10 to the same colormap of Fig.6 and Fig.7.

Wrong figure in Fig. 10 (d) is replaced.; discussion in the manuscript is also changed. P14 L399-401.

Authors' response to RC1

General comments:

This study assessed the application of different viewing geometries for a pair of geostationary imagers, AHI-AGRI and AHI-AMI to retrieve aerosol top height (ATH) information. The stereoscopic algorithm is presented, which converts the lofted aerosol layer parallax, calculated using image-matching of two visible images, to ATH. What is strongly missing in the manuscript is a discussion on the required ATH quality for different applications. I am not an expert in utilizing the ATH data but knowing the PBL processes I assume that 1-2km offset between calculated and measured on the ground or retrieved from CALIOP ATH is too high and further improvements of the retrieval approach are needed to produce a product of the required quality. The insufficient quality of a product is a cause for my decision to reconsider the manuscript after major revision, when a better (required for certain applications) quality of the product has been achieved.

We appreciate the reviewer's comments. It has been very difficult and thus rare to have ATH information from satellite imaging instruments which would provide valuable dataset over wider area. As suggested by the reviewer, detailed PBL process studies require higher accuracy ATH information. However, ATH with lower accuracy would still provide valuable information in understanding the ATH of long-range transport, the conversion of columnar aerosol optical depth (AOD) to surface PM concentrations, and tracking of wildfire and dust outbreak aerosols etc.

Nanda et al. (2020)-AMT used hyperspectral observation from TROPOMI to retrieve aerosol layer height. The results are compared to CALIOP extinction weighted mean height from 1 May 2018 to 28 February 2019. The mean bias was -2.41 km for land, and -1.03 km for the ocean. The standard deviation was 3.56 km for land, and 1.97 km for the ocean. Lee et al. (2021)-IEEE used VIIRS, OMPS (passive), and CALIOP (active) data for simultaneous retrieval of aerosol scattering property and aerosol layer height. The retrievals were conducted only over wildfire smoke layers over 42 cases from 2012 to 2018. Using only passive sensors (VIIRS, OMPS), the result showed a mean bias of -0.1 km and RMSE of 1.1 km compared to CALIOP extinction weighted mean height. Chen et al. (2021)-RSE also used TROPOMI but a different wavelength from Nanda et al. to retrieve absorbing aerosol (smoke, dust) height. This study compared retrieved aerosol height with CALIOP extinction weighted mean height at over 5 smoke cases and 2 dust cases. The results showed a mean bias of -0.01 km and RMSE of 0.64 km.

To give insights into the retrieval quality from our algorithm, we conducted error analyses of AHI-AGRI ATH. Please note that the retrieval algorithm is changed. We generated two heights from the CALIOP profile. One is called "90 % extinction height", which was used in our manuscript. The other is "extinction weighted mean height" which was used over other studies using spectroscopic methods such as Nanda et al. and Chen et al.

$$\text{ext. weighted mean height} = \frac{\sum_{i=1}^{n} \beta_{ext,i} Z_i}{\sum_{i=1}^{n} \beta_{ext,i}}$$

where $\beta_{ext,i}$ is extinction coefficient at 532 nm at height index $i$ and $Z_i$ is the altitude at $i$.

[Figure]

**Figure AR1 Comparison with CALIOP 90 % extinction height (upper) and CALIOP extinction weighted mean height (lower) according to correlation coefficient of best correlated moving window during image-matching process. Red boxplots show 5th, 25th, 50th, 75th, 95th percentiles of (retrieved ATH – CALIOP height). Blue circles show RMSD of retrieved height and CALIOP height.**

Here, the "correlation coefficient" means the value of the best correlated moving window (from the matching image) for a fixed window (from the reference image). As shown in figure AR1, it can be seen that as a the correlation coefficient becomes higher (which means that the algorithm successfully found the same aerosol layer on the image), the retrieval bias gets close to 0, and RMSD decreases. We then compared error analysis from all correlation coefficient data and that from quality-controlled (correlation coefficient > 0.95) data.

[Figure]

**Figure AR2 Same as Fig. AR1. But according to CALIOP integrated aerosol extinction. Left panels are for all correlation coefficient values while right panels are only for correlation coefficient > 0.95.**

Error analyses according to aerosol loading (column integrated CALIOP L2 aerosol extinction coefficient). Using all data regardless of the correlation coefficient, retrieval quality increases as aerosol loading increases. In this case, RMSD with CALIOP 90% extinction height (extinction weighted mean

height) decreased from 3.21 (3.08) km to 2.07 (2.35) km. Meanwhile, using quality-controlled (correlation coefficient > 0.95) data, the total RMSD was 1.66 km. Therefore, we could say that our results are compatible with the other studies.

Additionally, comparing both 90 % extinction height and extinction weighted mean height to the stereoscopic ATH. We could also inductively say that the stereoscopic algorithm gives an altitude that is near the top of the aerosol profile.

Specific comments:

How is ATH defined in the study? How it differs from aerosol layer height? This question came to my mind on P8 L 232

We tried to explain what our algorithm would give in the last paragraph on Chap. 3.1, which however seems confusing because it is mixed up with the definition in terms of height-parallax conversion. We moved the definition of ATH that is formed by the height-parallax conversion process to the paragraph above (P 6 L164-168). Then we described how the products are going to work in different situations (e.g., dense aerosol plume, thinner aerosol layer, multiple layers of aerosol) in P6 L169-184)

P2, L 31. How narrow? Please, provide numbers or refer here to Sect. 2.2.1

According to Winker et al. (2010, BAMS), CALIOP has 70 m footprint diameter.; P2 L34-35

P2, L 31. Please, specify more exactly bypasses time

With the word "bypass", we meant the active sensors missing the aerosol transport events. Sorry for the misleading. We changed the sentence to "active sensors such as CALIOP have very narrow swath (e.g., CALIOP footprint diameter is 70 m; Winker et al., 2010), which means that they may miss aerosol transport events most of the time.".; P2 L34-35

P2, L 35. I suggest using the word "distribution" instead of "structure"

Done. Thanks.

P2, L 59. Remove "data"

Done. Thanks.

P2, L 62-64. Is it lack if channels or lack of the stereoscopic view, which is insufficient?

We meant lack of channels that are sensitive to the height of the aerosol layer. For the sake of clarity, we changed the sentence to "the visible to infrared (VIS–IR) wavelength channels that are usually employed by meteorological satellite instruments usually lack sensitivity to aerosol height information, thus insufficient for the retrieval of aerosol height from observed radiances."; P3 L64-66

P4, Sect 2.1.2. Please, add bands characteristics, as in 2.1.1

Thank you for the suggestion, detailed band characteristics like 2.1.1 seem better to understand. We added similar sentences about observation bands to 2.1.2 "AMI also has 16 spectral bands, including 3 VIS, 1 NIR, 2 shortwave IR, and 10 IR channels. Blue and green bands (0.47, 0.51 μm) have spatial resolutions of 1 km at the sub-satellite point, and a red band has 0.5 km resolution (0.64 μm)."; P4 L104-106

P4, L 96. The Advanced Meteorological Imager (AMI) ….

Done. Thanks.

P4, L 97. Please specify new channels if they are used in the study. If not, it is not necessarily to

mentioned added channels here.

We appreciate the suggestion; we guess mentioning the new channels is not necessary for this paper. We simplified the sentence to "The Advanced Meteorological Imager (AMI) is a GEO meteorological instrument onboard Geo-KOMPSAT 2A (GK-2A), which was launched on 4 December 2018 by the National Meteorological Satellite Center (NMSC) of Korea succeeding the mission of its MI predecessor.".; P4 L 102-104

P4, L 99. Please, clarify: The AMI spectral bands are similar to those of AHI, except for a VIS and IR band; the center wavelengths and spatial resolutions of the VIS bands of AMI and AHI are similar.

Added band characteristics and deleted the unclear sentences.; P4 L104-106

P4, L 117-118. Please, rephrase

Done; P4 L123-124

P4, L 120. CALIOP product is less accurate compared to the ground-based measurements. I suggest naming of the inter-comparison with CALIOP as evaluation, instead of validation. I also suggest discussing first an opportunity for validation with ground instruments and second mention the evaluation with satellites (which in general have an advantage in coverage, though CALIOP coverage is quite small, but may allow evaluation in the conditions where ground instruments are missing)

We should've been careful in using "validation" when it comes to the aerosol height retrieval, thank you. We changed the word to "evaluation". P4 L124

We agree that long-term validation with ground-based lidar data would help demonstrate the feasibility of the stereoscopic aerosol height retrieval algorithm. Unfortunately, for the period from 1 January 2020 to 30 April 2020, only 49 days are collocated within 5 km from the lidar site. According to a conversation with Dr. Yeo, who provided us with the lidar data of SNU and GSN sites, the lidar signal would be totally dissipated when a thick aerosol layer is present. This indicates that a favorable condition for stereoscopic aerosol height retrieval algorithm is not the case for ground-based lidars. Also, even though the ground-based observation system works automatically, it needs manual maintenance from time to time, which leads to fewer data availability.

The objective of the comparison with ground-based lidar data is to show the possibility to monitor diurnal variation of aerosol height using geostationary passive sensors. Since many studies that used LEO satellites cannot monitor the hourly variation of aerosol vertical features, it is one of the strengths of the stereoscopic aerosol height retrieval algorithm using GEO satellites. We notice the need to clarify the purpose of comparison with ground-based lidar. Therefore, we put additional discussion as follows in P13 L 381-383.

P6, Sect. 3.1 Have you considered to develop two approaches, one for land and one for ocean, to resolve the ocean/land contribution at different wave lengths?

We did use different wavelengths to test our algorithm. On May 12, 2020, a thick dust plume was transported toward the Korean peninsula, which is a very favorable condition for a stereoscopic algorithm to work. Fig. AR3 shows image-matching correlation coefficients using 0.4, 0.6, and 0.8-micron channels to retrieve aerosol height. First, using a 0.4-micron channel, matching correlation values that find the same aerosol feature is low. A low correlation coefficient is expected over the ocean because the ocean is brighter at shorter wavelengths, but it was the same for the land. As shown in a single channel image in Fig. AR4, the surface seems darker at 0.4-micron but spatial patterns over the surface are more obvious at the channel too. This can be the reason why the correlation coefficient is lower than the 0.6-micron channel. For the 0.8-micron channel, correlation coefficient results seem okay.

But as shown in a single channel image in Fig. AR4-c, the aerosol layer is brighter at 0.6-um channel. For this kind of thick aerosol plume, the 0.8-micron retrieval may work, but when it comes to lower aerosol loading, the 0.8-micron may lack sensitivity. Therefore, we fixed the algorithm to use a single 0.6-micron channel.

[Figure]

**Figure AR3 Correlation coefficient of the best correlated moving window of 0.4 (a), 0.6 (b), and 0.8 (c) $\mu$m channels. \*more spatial coverage of (b) is due to algorithm change of AOD cutoff from 0.6 to 0.3.**

[Figure]

**Figure AR4 Single channel images of 0.4 (a), 0.6 (b), and 0.8 (c) $\mu$m channels.**

P6, L 160. Please, start with the definition of the parallax, then continue with the description of how is was calculated.

Thank you for your suggestion, we moved the definition of parallax to the front of the paragraph.; P6 L164-167.

P6, L 167. Please, provide short definitions here

The sentence is removed during revision and examples about other aerosol height retrieval algorithms can be found in Sect. 1.

P7, L 191. Based on what the AOD lower limit of 0.6 was chosen?

Since it was a feasibility study for stereoscopic aerosol height retrieval, we set the lower limit of AOD as 0.6 to get more robust results. We tested stereoscopic retrievals over pixels with AOD > 0.3. Through error analysis shown in Fig. AR1 and AR2, we showed that using data with the best matching correlation coefficient > 0.95, robust results were found regardless of aerosol loading. So, we changed the AOD lower limit from 0.6 to 0.3.

P8, L 228. Based on what the limit of 10km for the ALH was chosen? Can all pixels in the moving window be checked on the presence of AOD data? This will allow avoiding the influence of clouds.

The study area is a region where an aloft aerosol layer is observed with dust transport, which does not exceed 10 km usually. But the cutoff altitude of 10 km is still not physically reasonable because the aerosol layer can float over 10 km altitude during heavy smoke plume events and/or volcanic eruption.

For the sake of algorithm robustness, we removed the procedure. Also, thanks to your suggestion, we

changed some parts of our algorithm as you mentioned. Moving windows that have more than 20% of total pixels identified as a cloud by AHI AOD were removed from the image-matching process. Fig. AR5 shows the result of simply discarding ATH over 10 km on the left and the result of the new algorithm on the right. We now see a more reasonable result with less cloud contamination. P7 L202-205 describes corresponding changes in the algorithm.

[Figure]

**Figure AR5 ATH map of over-10 km-cutoff (a) and moving window cloud detection (b) on 4th April 2020.**

P9, L 242. What is INR error? Is it calculated based in instrument specifications?

To evaluate the uncertainty caused by wrong grid registration, we moved all pixels of the AHI image by 1 km. This simulates INR error from a satellite. Although it is not calculated based on the instrument specification (pointing accuracy and stability), the evaluation of the uncertainty regards instrument specifications, wrote as "considering the actual INR errors of the satellites (approximately 0.5, 1, and 4 km at channels with 1 km resolution for AHI, AMI, and AGRI, respectively), the INR error would not be of concern for the retrieval of aerosol heights of a few kilometers.". To clarify, we changed the expression "INR error" to "INR shift" and rephrased P9 L247.

P9, L 273. Agree. Why "a simple cause of retrieval uncertainty was involved here?" (L 242)

The retrieval performance of the stereoscopic retrieval algorithm can be estimated by evaluating how accurately the algorithm calculates the parallax. Since a false location of the image is the biggest possible error source, only the INR error is considered. Still, minor errors can be introduced due to surface signal intrusion. However, quantitative computation of how much these sources affect the parallax calculation is impossible. So, we wrote as "a simple cause of retrieval uncertainty (which means the INR error) was involved here". It seems to be a confusing phrase, so we changed it to "Since false registration of satellite grid introduces error on parallax calculation, uncertainty from satellite INR error needs to be calculated."; P9 L248-250

P10, L 302. Please add the definition for EC

Done; P11 L315-316

P11, L 308. Please, replace "valid" with "retrieved" or "provided".

Done; P11 L321

P11, L 309. Please, replace "valid" with "retrieved" or "calculated".

Done; P11 L322

P11, L 306-309. Please add to the text the reference to Fig. 6a and 6b .

Thank you for pointing out, we added references of Fig. 6a, b and Fig. 7a, b to the relevant sentences.

P11, L 310. The difference of 2 km in the ATH estimation is significant, when one think about the

location of aerosol layer regarding the planetary boundary layer (PBL). The knowledge on that (within or above PBL) is important for predicting the further aerosol transport directions and intensity. This is more critical for high AOD loading episodes, which you consider. Why 1 and 2km difference was chosen as criteria for evaluation? This is very big offset, if we think about possible applications of the calculated ATH. What are typical criteria for CALIOP ATH evaluation? Other ATH products?

There are a few studies that assessed the retrieval uncertainty of aerosol height from satellite remote sensing. For example, Lee et al. (2015) discussed uncertainties from individual error sources such as AOD, SSA, surface elevation, …, and concluded that the uncertainty in the retrieved aerosol height is estimated from -1.20 to 1.80 km over land and from -1.15 to 1.58 km over the ocean when favorable conditions are met. Therefore, 1 and 2 km difference was chosen based on the uncertainty assessment from previous work of Lee et al.

P11, L 315-331. Can you discuss the conditions in which the disagreement between two products is most pronounced? And provide plot for AHI-AGRI vs AHI-AMI ATH.

Fig. AR6 shows a 2-dimensional histogram plot of AHI-AGRI vs AHI-AMI ATH. Due to the lack of sensitivity for AHI-AMI pair, the result is very scattered. Comparing with CALIOP (Fig. 6 and 7 in the manuscript), we discussed that since the distance between AHI and AMI is too close for stereographic aerosol feature retrieval, result of the pair is erroneous in any conditions. Therefore, we concluded that AHI-AMI ATH is of no use.

[Figure]

**Figure AR6 2-dimensional histogram of AHI-AGRI ATH vs. AHI-AMI ATH.**

P12, L 342 overestimated…. or aerosols were distributed evenly along the height

For the word "overestimated", we wanted to say that AHI AOD was overestimated. However, what we meant was that the retrieval error is due to small CALIOP EC53 values. It seems misleading, so we changed the sentence to "Unlike the first case, the CALIOP EC profile of the latter case has few values of $> 0.3 \text{ cm}^{-1}$, indicating retrieval error caused by low aerosol loading."; P12 L351-352

P13, Sect.5. To my understanding, collocation of geostationary satellite with ground measurements provides an opportunity for considerably higher number of collocations than with CALIPSO. However, only two cases are considered. To make a conclusion on the validation results, statistics (bias!) should be calculated using all possible collocations. Scatter plot as Fig.8 as well as frequency distribution plot are needed to be presented and discussed.

Please find the limitation of using ground-based lidar data for validation in response to P4 L120.

P13, L 397. Why AOD limit of 0.6 was applied, if "…. not affected by variations in aerosol….

It seems misleading. We changed the sentence to "Furthermore, the method is not affected by variations in aerosol optical properties when the image-matching method is successfully worked.". Please refer to the supplement figures and related error analysis at P14 L 410.

Figure 4. Please, check the location of AMI

Done. Thanks.

Figure 6 c,d. Please change the color scale to see better the difference between two pairs.

Done. Thanks.

<References>

Chen, X., Wang, J., Xu, X. G., Zhou, M., Zhang, H. X., Garcia, L. C., Colarco, P. R., Janz, S. J., Yorks, J., McGill, M., Reid, J. S., de Graaf, M., and Kondragunta, S.: First retrieval of absorbing aerosol height over dark target using TROPOMI oxygen B band: Algorithm development and application for surface particulate matter estimates, Rem. Sens. Environ., 265, ARTN 112674, doi:10.1016/j.rse.2021.112674, 2021.

Lee, J. H., Hsu, N. C., Sayer, A. M., Seftor, C. J., and Kim, W. V.: Aerosol layer height with enhanced spectral coverage achieved by synergy between VIIRS and OMPS-NM measurements, IEEE Geosci. Rem. Sens., 18, 949–953, doi:10.1109/Lgrs.2020.2992099, 2021.

Nanda, S., de Graaf, M., Veefkind, J. P., Sneep, M., ter Linden, M., Sun, J. Y. T., and Levelt, P. F.: A first comparison of TROPOMI aerosol layer height (ALH) to CALIOP data, Atmos. Meas. Tech., 13, 3043–3059, doi:10.5194/amt-13-3043-2020, 2020.

Winker, D. M., Pelon, J., Coakley, Jr. J. A., Ackerman, S. A., Charlson, R. J., Colarco, P. R., Flamant, P., Fu, Q., Hoff, R. M., Kittaka, C., Kubar, T. K., le Treut, H., McCormick, M. P., Mégie, G., Poole, L., Powell, K., Trepte, C., Vaughan, M. A. and Wielicki B. A.: The CALIPSO Mission: A Global 3D View of Aerosols and Clouds, Bull. Ameri. Meterol. Soc., 1211-1229, 2010.

Authors' response to RC2

We would like to thank the Reviewer for his/her thorough and detailed review as well as for the suggested papers. Our responses (in blue) for each comment (in black) are provided below.

**[updates in the algorithm]**

Lowered the retrieval cutoff AHI AOD from 0.6 to 0.3.

Removed 10 km ATH cutoff.

Image-matching is no more conducted when 20 % of the pixels in moving (or, reference) window is cloud-contaminated.

**[changes in figures]**

Figures use updated version of ATH.

CALIOP L2 aerosol extinction profile is replaced to CALIOP L1 total attenuated backscatter in Fig.6 (e) and Fig. 7 (e).

Fig. 8 is replaced with 2-dimensional histogram.

Added x=0 line in Fig. 9.

Changed colormaps in Fig. 10 to the same colormap of Fig.6 and Fig.7.

Wrong figure in Fig. 10 (d) is replaced.; discussion in the manuscript is also changed.; Lines399-401.

Summary: This paper explores the utility of stereoscopic methods from geostationary satellites to make an aerosol plume height estimate. With the advent of next generation imagers over Asia (GK-2A; Himiwari-8, FY-4 etc.), North/South America (GOES 16, 17, and now 18), and now starting in Europe (MTG), there is great potential in combining data in overlap zones. This work is one of several in recent years trying to make use of this new capability. Here, the authors provide a relative brief paper on their experiments combining AHI with AMI and AGRI to derive aerosol heights for cases around the Korean Peninsula. The paper briefly gives a rationale for the work, a list of data sources, explication of geometry, and finally performs comparisons to CALIOP and the Korean Lidar Network. As one would expect, skill is favored by geometries with wider separation between instruments.

After reading the paper, it was not clear to me how well the products actually work. In concept (as the authors note) there are fundamental differences in what is produced between spectroscopic aerosol height methods (say OA&B based TROPOMI based) and stereographic assessment (e.g. MISR). Spectroscopic methods give a centroid height, and stereographic give a feature height. The way the paper is laid out, it is not clear what specific features are being keyed off on in the algorithm. For clouds it can be straightforward, for aerosol features uncles there is a dense plume there are multiple textures in the satellite imagery. There is no discussion on this, and other assumptions such as the presence of embedded clouds they admitted they ignored. Looking at the scatter plots, there is so much scatter I am not sure their algorithm beats a climatology. E.g., if one takes the average of the average of CALIOP heights, does the retrieval's RMSD beat that? And if so by how much? It would not surprise me that there is a bias, but what we really want to know is if the deviations as observed by the retrieval match CALIOP? Further their cases are all pretty close to the surface. Perhaps add a Siberian smoke plume case?

Anyways, while I think this is a fine effort, the paper lacks details that are required and thus I suggest major revisions. I would encourage the authors add a few more pages too zooming in further on the case studies so we can see what the retrieval is looking at. Maybe a figure that pulls the string through that includes a higher dynamic range of features? Best wishes with the endeavor.

[General response]

Thanks for the thorough review comments. We added discussion about how the products work in different situations (dense aerosol plume, thinner aerosol layer, multiple layers of aerosol).; Line 165-180. For the most favorable condition, the algorithm is likely to give the height near the top of the aerosol layer, we define what the algorithm gives as aerosol top height.

About the embedded clouds, we changed the algorithm to be less affected by the embedded (or nearby) clouds by removing pixels with no AHI AOD values. So, the discussion about the embedded clouds seems to be treated as an error source when AHI AOD fails to screen out cloud pixels. Therefore, we mentioned the effect of embedded clouds in Section 3.3 with an additional cloud screening process in the current revision.; Line 202-205.

To give insights into the retrieval quality from our algorithm, we conducted error analyses of AHI-AGRI ATH. Please note that the retrieval algorithm is changed. We generated two heights from the CALIOP profile. One is called "90 % extinction height", which was used in our manuscript. The other is "extinction weighted mean height" which was used over other studies using spectroscopic methods such as Nanda et al. and Chen et al (2021).

$$\text{ext. weighted mean height} = \frac{\sum_{i=1}^{n} \beta_{ext,i} Z_i}{\sum_{i=1}^{n} \beta_{ext,i}}$$

where $\beta_{ext,i}$ is extinction coefficient at 532 nm at height index $i$ and $Z_i$ is the altitude at $i$.

[Figure]

**Figure AR4 Comparison with CALIOP 90 % extinction height (upper) and CALIOP extinction weighted mean height (lower) according to correlation coefficient of best correlated moving window during image-matching process. Red boxplots show 5th, 25th, 50th, 75th, 95th percentiles of (retrieved ATH – CALIOP height). Blue circles show RMSD of retrieved height and CALIOP height.**

Here, the "correlation coefficient" means the value of the best correlated moving window (from the matching image) for a fixed window (from the reference image). As shown in figure AR1, it can be seen that as the correlation coefficient becomes higher (which means that the algorithm successfully found the same aerosol layer on the image), the retrieval bias gets close to 0, and RMSD decreases. We then

compared error analysis from all correlation coefficient data and that from quality-controlled (correlation coefficient > 0.95) data.

[Figure]

**Figure AR2 Same as Fig. AR1. But according to CALIOP integrated aerosol extinction. Left panels are for all correlation coefficient values while right panels are only for correlation coefficient > 0.95.**

Error analyses according to aerosol loading (column integrated CALIOP L2 aerosol extinction coefficient). Using all data regardless of the correlation coefficient, retrieval quality increases as aerosol loading increases. In this case, RMSD with CALIOP 90% extinction height (extinction weighted mean height) decreased from 3.21 (3.08) km to 2.07 (2.35) km. Meanwhile, using quality-controlled (correlation coefficient > 0.95) data, the total RMSD was 1.66 km. Therefore, we could say that our results are compatible with other studies.

Additionally, comparing both 90 % extinction height and extinction weighted mean height to the stereoscopic ATH. We could also inductively say that the stereoscopic algorithm gives an altitude that is near the top of the aerosol layer.

[Figure]

**Figure AR3 2-dimensional histogram of CALIOP 90% extinction height and stereoscopic ATH of AHI-AGRI (a) and AHI-AMI (b).**

The scatter plots in Fig. 8 are "scene-averaged" data. We chose this because it was hard to read the plot when we draw all collocated pixels. However, averaging all the collocated values of ATH and CALIOP height can be affected by outliers. Therefore, we changed Figure 8 to 2-dimensional histograms using quality-controlled retrieval. % within 1 km from CALIOP 90% extinction height is 57.4% and % within 2 km from CALIOP 90% extinction height is 88.9% for AHI-AGRI. The same for AHI-AMI is 5.9% and 24.4%.

Line 49: "Studies have shown that the use of geometrical features of elevated atmospheric structures apparent to multiple sensor imagery is effective, rather than using computationally expensive radiative transfer calculations." Well it is more than that. Spectroscopic techniques give a different product altogether-the centroid of a plume- and that is compounded when one has multiple aerosol layers. Stereography gives a plume top for those cases when one can see a feature. This is a very different thing. In practice, I would say plume height from say MISR is more tractable than the spectroscopic methods.

We appreciate sharing your insights about the retrieved height from different approaches. We added what information spectrographic algorithms get to retrieve from observation data as "Using stereography, unlike spectroscopic algorithms, one gets to retrieve feature top height." in line 53-54.

Line 72: "Cloud top heights have been successfully retrieved using geometrical fusion of two geostationary satellite images (Lee at al., 2020), suggesting the applicability of such a method to any structures in the atmosphere." A quick look on web of science I found several papers on the topic of stereo heights for aerosol features worth mentioning-some of which have a lot of parallels to this paper, including Lee et al (2020)-Remote Sensing; Merucci et al., (2016)-Remote Sensing.; Prata and Lynch (2019)-Atmosphere just to name a few. And for clouds there are many more. So the field is more advanced than the paper is letting on.

Thank you for your suggestions on additional references. We added other references (Hasler, 1981; Seiz et al., 2007; Zašek et al., 2013; Merucci et al., 2016) that showed LEO/GEO stereoscopic ash/metrological cloud top height retrievals.; Line 76

Line 73: "However, aerosol layers are not as optically thick as clouds and their heights are much lower than cloud tops, so the applicability and accuracy of the geometrical method for estimating ATH remain unresolved" I am not sure what you mean by using the word "unresolved" here. You can say that about anything really. You can say it is in the early stage of development. But given the complexity of the system it may never be "resolved"

What we tried to deliver with the word "unresolved" was that since the previous studies are focused on clouds or volcanic ash plumes, the application of the geometrical feature height retrieval needs to be tested for aerosol layers in East Asia, where aerosol layers are formed due to surface pollution emission thus much thinner/lower than clouds or volcanic ashes. Sorry for the confusion. We changed the sentence to "However, typical aerosol layers are formed due to surface pollution emission in East Asia thus are not as optically thick as clouds or volcanic ash plumes. Also, aerosol layers tend to be at a much lower height than cloud or volcanic ash plumes. So, the applicability and accuracy of the geometrical method for estimating aerosol feature height needs to be investigated".; Line77-80

Line 117: "Their ability of lidar observations to produce aerosol profile data with high vertical resolution enables them to be a validation standard for spaceborne aerosol height retrieval algorithms." I think you mean the? Keep in mind, it is best not to use pronouns in the first sentence of a paragraph. It is often unclear what "they, them, it" refers to.

Thank you for pointing out the ambiguity. We rephrased it as "Through intercomparison with aerosol profile data from lidars, spaceborne aerosol height retrieval algorithms with passive sensors can be evaluated.".; Lines 123-124

Line 153-Here you reference Lee et al., 2020, but I don't think it is in the references.

Sorry for the confusing references. Since the fact that the stereoscopic ATH retrieval is based on other LEO/GEO cloud/volcanic ash top height retrieval algorithms is shown in Section 1., we deleted the sentence.; Line 157

Line 225: "Although highly reflective clouds can interfere with correlation calculations, the stereoscopic ATH algorithm does not include a cloud masking procedure. By using the AHI AOD product, where retrievals are undertaken only for cloud-free pixels, we assume that selected pixels with high AOD are cloud free." Is that a good assumption? So also what happens when you have a thick smoke or dust plume? If the retrieval fails under these circumstances some of the strongest aerosol features may be missed. In future versions, the authors may want to think hard about the aerosol-cloud discrimination.

We are aware of the aerosol-cloud discrimination that follows when using AHI AOD data. Considering that this study aims to examine the feasibility of stereographic atmospheric feature retrieval method to aerosol layers, developing an internal could-aerosol discrimination procedure is of too much complexity. Since we got reasonable results with the pair of AHI and AGRI, we are planning to further advance our algorithm containing internal high aerosol loading detection.

Line 292: It is unclear to me how the CALIOP heights should be generated. The discussion of CALIOP in this and the data section is brief. To do a proper match up, CALIOP needs to observe a feature, which has high enough AOD and texture for the matching algorithm to identify it as a target. CALIOP may be able to give the AOD, but what about the texture feature? This is not really described in the paper, but is the lynchpin problem. For the discussion with Lee 2021, they had to do a extinction weighted mean to compare apples to apples. For this paper the CALIPSO baseline "We therefore define CALIOP height as the height where the cumulative EC532 represents 90 % of the total column integrated EC532, starting from the bottom of the profile." But this has no feature identification. If you have a plume embedded in a polluted atmosphere you could be looking at completely different things.

We agree with you. For the proper CALIOP height match-up, it would be best to find the altitude that has enough aerosol loading and shows texture for the image-matching to work. However, searching for horizontal texture in the CALIOP profile data was challenging because it gives only 1-axis for the horizontal plane (narrow footprint of ~70m; Winker et al. 2010). The reason why we set the CALIOP height as "the height where the cumulative EC532 represents 90 % of the total column integrated EC532, starting from the bottom of the profile." is because we defined the retrieved stereoscopic height as aerosol top height (please, refer to the general response). For the methodology for the definition of CALIOP profile top height, we referred to Lee et al. (2015).

To check if different definition of CALIOP height would be better for the comparison with stereoscopic ATH, we also conducted an error analysis with extinction weighted mean height (Fig.AR1, AR2). The results show that compared to the CALIOP extinction weighted mean height, stereoscopic aerosol height has a systematic positive bias. Therefore, we inductively concluded that the definition of CALIOP height as "the height where the cumulative EC532 represents 90 % of the total column integrated EC532" is reasonable.

However, we see that the discussion on the definition of CALIOP height is too brief. So, we mentioned the difficulty of getting horizontal texture of aerosol layer from CALIOP data. Also, the word "CALIOP

height" seems ambiguous. So, we added more explanation and changed the expression "CALIOP height" to "CALIOP 90% extinction height". Please find the revised paragraph on lines 310-317.

Line 306: Why were these cases picked? Perhaps you should pick some with more dynamic range, like a Siberian smoke plume

 The case of Fig. 6 (23 January 2020) represents a case of an aerosol layer that is out of retrieval sensitivity for the AHI-AMI pair but is retrievable with AHI-AGRI. We tried to show a case of a higher aerosol layer with Fig.7 (8 April 2020) which had an aerosol layer reaching 6 km altitude. We picked cases where CALIOP flew over the aerosol layer. Unfortunately, we couldn't find Siberian smoke plumes having valid CALIOP profile data within the study area. For the Siberian smoke plumes out of the area of interest, they were frequently found over 65°N, where AHI AOD cannot retrieve AOD because of high solar zenith angle. Therefore, we tried to pick a scene and retrieve stereographic aerosol height regardless of AOD values. However, coarse spatial resolution near the edge of images made the resampling process AGRI (or, AMI) pixels farther than 5 km from AHI. This made a too much-smoothed image. As a result, the image-matching process couldn't work properly.

We added more discussion in Sect 4.1, where we mentioned briefly the coarser resolution making retrieval unfavorable (lines 272-274), to clarify why we set the study region as East Asia. Additionally, we added characteristics of aerosols in East Asia that the stereographic algorithm focused on during development. The updated discussions can be found on the lines 77-80.

Line 345: Looking at the scatter plots, it is not clear to me what skill this method even has for the cases they are investigating.   If one assumes a simple baseline from climatology (say the mean from CALIOP or maybe 3.5 or 4 km), does this method provide any skill beyond that?   Here I am more worried about the RMSD than the bias.   Anything systematic you can say about what situations it works and when it doesn't overall? This really needs to be added to the paper.

As shown in the general response, we conducted an error analysis and calculated RMSD for CALIOP 90 % extinction height. The error analysis for AHI-AMI is also done and we want to provide the results as supplement figures (Fig. AR4). Also, the scatter plots that show "scene-averaged" height are replaced with a 2-dimensional histogram of all collocated points. Unfortunately, a few points show ATH over 5 km. However, studies about CALIOP aerosol height climatology in East Asia (Liu et al. 2019; Gui et al. 2022) show that aerosol layers in East Asia are usually under 5 km. We are planning to broaden the study area towards South Asia, where elevated biomass burning smoke is observed.

[Figure]

**Figure AR4 Same as right panels of Fig. AR2. But showing error analysis of AHI-AGRI ATH (a) and AHI-AMI (b).**

Line 370: I think an error distribution like was done with CALIOP would help synthesize your findings.

We agree that long-term validation with ground-based lidar data would help demonstrate the feasibility of the stereographic aerosol height retrieval algorithm. Unfortunately, for the period from 1 January 2020 to 30 April 2020, only 49 days are collocated within 5 km from the lidar site. According to a conversation with Dr. Yeo, who provided us with the lidar data of SNU and GSN sites, the lidar signal would be totally dissipated when a thick aerosol layer is present. This indicates that a favorable condition for stereoscopic aerosol height retrieval algorithm is not the case for ground-based lidars. Also, the ground-based observation system needs manual maintenance, which leads to fewer data availability of ground-based lidar.

The objective of the comparison with ground-based lidar data is to show the possibility to monitor diurnal variation of aerosol height using geostationary passive sensors. Since many studies that used LEO satellites cannot monitor the hourly variation of aerosol vertical features, it is one of the strengths of the stereoscopic aerosol height retrieval algorithm using GEO satellites. We notice the need to clarify the purpose of comparison with ground-based lidar. Therefore, we put additional discussion as follows in lines 381-383.

<References>

Chen, X., Wang, J., Xu, X. G., Zhou, M., Zhang, H. X., Garcia, L. C., Colarco, P. R., Janz, S. J., Yorks, J., McGill, M., Reid, J. S., de Graaf, M., and Kondragunta, S.: First retrieval of absorbing aerosol height over dark target using TROPOMI oxygen B band: Algorithm development and application for surface particulate matter estimates, Rem. Sens. Environ., 265, ARTN 112674, doi:10.1016/j.rse.2021.112674, 2021.

Gui, L., Tao, M., Wang, Y., Wang, L., Chen, L., Lin, C., Tao, J., Wang, J., and Yu, C.: Climatology of

aerosol types and their vertical distribution over East Asia based on CALIPSO lidar measurements., Int. J. Climatol., 1-13, doi:10.1002/joc.7599, 2022.

Hasler, A. F.: Stereographic observations from geosynchronous satellites: An important new tool for the atmospheric sciences., Bull. Am. Meteorol. Soc., 62, 194–212, 1981.

Lee, J., Hsu, N. C., Bettenhausen, C., Sayer, A. M., Seftor, C. J., and Jeong, M. J.: Retrieving the height of smoke and dust aerosols by synergistic use of VIIRS, OMPS, and CALIOP observations, J. Geophys. Res. Atmos., 120, 8372–8388, doi:10.1002/2015jd023567, 2015.

Liu, D., Zhao, T., Boiyo, R., Chen, S., Lu, Z., Wu, Y., and Zhao, Y.: Vertical Structures of Dust Aerosols over East Asia Based on CALIPSO Retrievals., Rem. Sens., 11, 701, doi:10.3390/rs11060701, 2019.

Merucci, L., Zakšek, K., Carboni, E., and Corradini, S.: Stereoscopic estimation of volcanic ash cloud-top height from two geostationary satellites. Remote Sens., 8, 206, 2016.

Nanda, S., de Graaf, M., Veefkind, J. P., Sneep, M., ter Linden, M., Sun, J. Y. T., and Levelt, P. F.: A first comparison of TROPOMI aerosol layer height (ALH) to CALIOP data, Atmos. Meas. Tech., 13, 3043–3059, doi:10.5194/amt-13-3043-2020, 2020.

Seiz, G., Tjemkes, S., and Watts, P.: Multiview cloud-top height and wind retrieval with photogrammetric methods:Application to Meteosat-8 HRV observations., J. Appl. Meteorol. Clim., 46, 1182–1195, 2007.

Winker, D. M., Pelon, J., Coakley, Jr. J. A., Ackerman, S. A., Charlson, R. J., Colarco, P. R., Flamant, P., Fu, Q., Hoff, R. M., Kittaka, C., Kubar, T. K., le Treut, H., McCormick, M. P., Mégie, G., Poole, L., Powell, K., Trepte, C., Vaughan, M. A. and Wielicki B. A.: The CALIPSO Mission: A Global 3D View of Aerosols and Clouds, Bull. Ameri. Meterol. Soc., 1211-1229, 2010.

Zakšek, K., Hort, M., Zaletelj, J., and Langmann, B.: Monitoring volcanic ash cloud top height through simultaneous retrieval of optical data from polar orbiting and geostationary satellites. Atmos. Chem. Phys., 13, 2589–2606, 2013.

---

## Referee Report (RR1)

Review for Atmospheric Measurement Techniques

Title: Exploring geometrical stereoscopic aerosol top height retrieval from geostationary satellite imagery in East Asia

By Minseok Kim, et al.

**General comments:**

This study assessed the application of different viewing geometries for a pair of geostationary imagers, AHI-AGRI and AHI-AMI to retrieve aerosol top height (ATH) information. The stereoscopic algorithm is presented, which converts the lofted aerosol layer parallax, calculated using image-matching of two visible images, to ATH.

During the reviewing process, the main following improvements have been done:

- The retrieval algorithm has been changed: Extinction weighted mean height has been generated.
- Error analysis has been conducted.

However, error analysis results (figs. AR1-AR4 + discussion) were not included into the manuscript.

Those changes allow considering the manuscript for publication in the AMT, after following comments have been addressed:

- including error analysis results into the manuscript
- thorough revision of the references. E.g., Kim et al., 2008, Lim et al., 2018, Zoogman et al., 2011, Merucci et al., 2016, ets., are missing in References.
- Last access date should be added to all web links (e.g., KALION, Line 155)

---

## Author Response (AR2)

**Reviewer 1:**

This study assessed the application of different viewing geometries for a pair of geostationary imagers, AHI-AGRI and AHI-AMI to retrieve aerosol top height (ATH) information. The stereoscopic algorithm is presented, which converts the lofted aerosol layer parallax, calculated using image-matching of two visible images, to ATH.

During the reviewing process, the main following improvements have been done:
- The retrieval algorithm has been changed: Extinction weighted mean height has been generated.
- Error analysis has been conducted.

However, error analysis results (figs. AR1-AR4 + discussion) were not included into the manuscript. Those changes allow considering the manuscript for publication in the AMT, after following comments have been addressed:

• **including error analysis results into the manuscript**

Thanks to your suggestion, we now included error analysis results and discussions in Section 5.1.2 "Long-term comparison with CALIOP" (L394-424). Also, the results are also mentioned in conclusions.

• **thorough revision of the references. E.g., Kim et al., 2008, Lim et al., 2018, Zoogman et al., 2011, Merucci et al., 2016, ets., are missing in References.**

Thanks for checking and sorry for the missing reference. References are corrected as follows:

  ✓ Published year of Zoogman et al. in the manuscript was corrected to 2017.

  ✓ Merucci et al., 2016 is added to the References.

  ✓ Kim et al., 2008 is added to the References.

• **Last access date should be added to all web links (e.g., KALION, Line 155)**
Last access date is added as (http://www.kalion.kr, last access: 29 March 2023).

last access date for NMSC INR report is added as (https://nmsc.kma.go.kr/homepage/html/satellite/quality/selectQualityGk2a.do, last access: 29 March 2023),

**Reviewer 2:**
The authors have clearly done quite a bit of work in response to the comments. We really appreciate the effort. However, looking at the scatter plots compared to the moving window correlation coefficient plots things are still unclear as to the skill of the retrieval. So in the moving window shows lower RMSD for higher correlation (which makes sense) but then in the regressions the skill does not look that good against what you would consider a baseline. I think the work here needs to be acknowledged, and it is clearly a hard problem. But I recommend publication a little bit more error analysis and in particular moving towards a prognostic error model would make this effort very useful to the community. **Can the authors please go into more details on what are the specific circumstances on when the algorithm works well versus when it doesn't?** Is it small clouds, or deeper layers, etc? This would be appreciated. Best wishes.

We understand that the quantitative prognostic uncertainty using such methods as model simulations would help demonstrating the algorithm's skill. However, considering that our

algorithm is based on parallax estimation from image-matching, quantitative assessment of uncertainty is limited. In Sect. 4.2 we showed a quantitative uncertainty induced by INR error, which makes systematic error. As the reviewer mentioned, this does not give the community information of specific circumstances on when the algorithm works well versus when it doesn't. Therefore, we conducted a quantitative assessment of uncertainty of some cases that may induce retrieval uncertainty (surface feature, and multiple layers of aerosol). This is done by simulating retrieval by generating two images from each satellite, and manually moving one image to get a parallax value that aerosol layer would make in real situation. Please note that impacts of small clouds are not examined because the algorithm works only on cloud-free images.

As shown in Fig. AR1, dense aerosol that blocks surface signal results in retrieval of the aerosol top height. However, stereoscopic ATH retrieval of thin aerosol layer that permits surface signals to penetrate results in image-matching of the surface (retrieved height = 0 km). Fig. AR2 shows cases of multiple layers. Addition of a thin upper aerosol layer does not change the retrieval results, but as the upper layers thicken, higher ATH (closer to the upper layer) is retrieved. Detailed discussions are in manuscript Sect 4.2.

[Figure]

**Figure AR1 A graphical illustration of stereoscopic ATH retrievals. Over (a) a surface with average albedo of 0.06, 2-dimensional Gaussian-shaped aerosol layers at 2.28 km with peak albedo of (b) 0.30 and (c) 0.10 are added.**

[Figure]

**Figure AR2 Same as Fig. AR1, but assuming a dark surface. Over (a) a single layer of aerosol with peak albedo of 0.10 at 2.28 km, three different upper aerosol layers at 4.57 km with peak albedo of (b) 0.03, (c) 0.10, and (d) 0.20 are added.**